# Data Selection for Language Models
# via Importance Resampling

**Sang Michael Xie, Shibani Santurkar, Tengyu Ma, Percy Liang**
Department of Computer Science
Stanford University
{xie, shibani, tengyuma, pliang}@cs.stanford.edu

## Abstract

Selecting a suitable pretraining dataset is crucial for both general-domain (e.g., GPT-3) and domain-specific (e.g., Codex) language models (LMs). We formalize this problem as selecting a subset of a large raw unlabeled dataset to match a desired target distribution given unlabeled target samples. Due to the scale and dimensionality of the raw text data, existing methods use simple heuristics or require human experts to manually curate data. Instead, we extend the classic importance resampling approach used in low-dimensions for LM data selection. We propose *Data Selection with Importance Resampling (DSIR)*, an efficient and scalable framework that estimates importance weights in a reduced feature space for tractability and selects data with importance resampling according to these weights. We instantiate the DSIR framework with hashed n-gram features for efficiency, enabling the selection of 100M documents from the full Pile dataset in 4.5 hours. To measure whether hashed n-gram features preserve the aspects of the data that are relevant to the target, we define *KL reduction*, a data metric that measures the proximity between the selected pretraining data and the target on some feature space. Across 8 data selection methods (including expert selection), KL reduction on hashed n-gram features highly correlates with average downstream accuracy ($r = 0.82$). When selecting data for continued pretraining on a specific domain, DSIR performs comparably to expert curation across 8 target distributions. When pretraining general-domain models (target is Wikipedia and books), DSIR improves over random selection and heuristic filtering baselines by 2–2.5% on the GLUE benchmark.[1]

## 1 Introduction

Given a fixed compute budget, the choice of pretraining data is critical for the performance of language models (LMs) [18; 10; 24; 64; 27]. Existing works rely on heuristics to select training data. For example, GPT-3 [10] and PaLM [14] filter web data for examples that are closer to formal text from Wikipedia and books as a proxy for high quality, a method which we call *heuristic classification*. Specifically, they train a binary classifier to discriminate formal text from web data and select web examples that have a predicted probability above a noisy threshold [10; 18; 21]. However, heuristic classification does not guarantee that the selected data is distributed like formal text. As a second example, domain-specific LMs such as Minerva [45] and Codex [11] (math and code LMs, respectively) employ domain-adaptive pretraining (DAPT) [24], where the model is initialized from a base LM and continues to be pretrained on a domain-specific dataset to achieve gains over the base LM on that domain. The domain-specific datasets are typically manually curated, but a framework for automating data selection could save effort and increase the amount of relevant training data.

---

[1]Code, selected data, and pretrained models are available at https://github.com/p-lambda/dsir.

37th Conference on Neural Information Processing Systems (NeurIPS 2023).

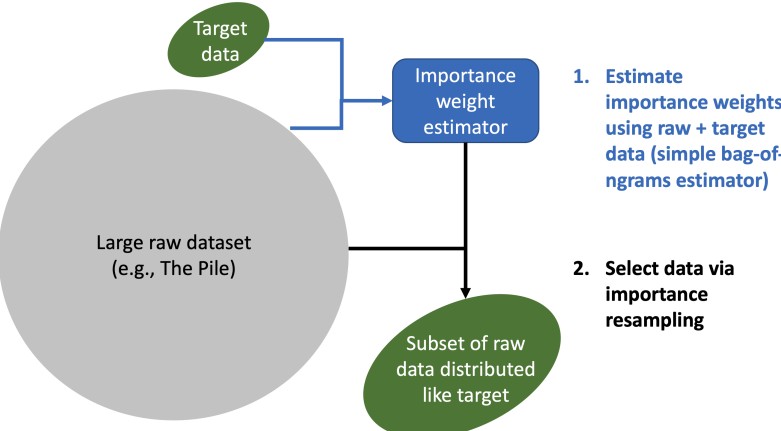

Figure 1: Given a large raw dataset such as The Pile [21] and a smaller target dataset (e.g., Wikipedia + books), we aim to select a subset of the raw data that is distributed like the target in some feature space. Our method, DSIR, first estimates importance weights using raw and target data in an n-gram feature space. The importance weights are used to resample a subset of the raw dataset.

In this paper, we consider the problem of **data selection**: given a large and diverse raw dataset (e.g., The Pile [21]) and a smaller dataset sampled from a desired target distribution, choose a subset of the raw data that is distributed similarly to the target (Figure 1). While a natural approach is to resample the raw data according to importance weights (importance resampling [68]), estimating importance weights on high dimensional data such as text is often statistically intractable [6; 75].

Instead, our **D**ata **S**election with **I**mportance **R**esampling (DSIR) framework efficiently estimates importance weights over a featurization of the raw and target distributions (Section 3). Our framework first maps the raw and target data onto some feature space and resamples a subset of raw data according to importance weights computed in this feature space. DSIR is extensible via the choice of feature space and importance estimator, which specify what aspects of the data we care about.

What is a feature space that both allows for efficient computation and preserves aspects of the data that are relevant for the target? In Section 4, we instantiate DSIR with hashed n-gram features, where n-grams are hashed onto a fixed number of buckets, for efficiency and scalability. The importance estimator is parameterized by bag-of-words generative models on the hashed n-grams, learned by simply counting the the hash bucket frequencies. DSIR with hashed n-grams enables the selection of 100M documents from the full Pile dataset in 4.5 hours.

To evaluate how well hashed n-grams preserves the aspects of the data that are relevant for the target, in Section 6 we define *KL reduction*, a data metric that measures how much a selected dataset reduces the Kullback-Leibler (KL) divergence to the target (in some feature space) over random selection ($\mathrm{KL}(\text{target}\|\text{random}) - \mathrm{KL}(\text{target}\|\text{selected})$). We show in Section 5 that KL reduction highly correlates with average downstream performance (Pearson $r = 0.82$) across 8 data selection methods, including expert selection.

We consider selecting data from The Pile (1.6B examples) for continued pretraining of domain-specific LMs and training general-domain LMs from scratch. First, we select data for continued pretraining [24] of domain-specific LMs in a controlled setting where the target samples are unlabeled training inputs from a downstream dataset (Section 5). We perform continued pretraining on the selected data starting from RoBERTa [48] and evaluate by fine-tuning on the downstream dataset (whose unlabeled inputs were also used as the target for data selection). On 8 datasets from 4 domains (CS papers, biomedical papers, news, reviews), DSIR improves over RoBERTa (no continued pretraining) by 2% on average and is even comparable to continued pretraining on expert-curated data from Gururangan et al. [24].

For general-domain LMs (Section 7), the data selection target is formal, clean text from Wikipedia and books, following GPT-3 [10]. We train a masked language model (MLM) from scratch on the selected data and evaluate by fine-tuning on GLUE [82]. In controlled experiments, heuristic classification performs comparably to random sampling from The Pile, possibly because The Pile is already filtered

using heuristic classification. DSIR improves over both baselines by 2–2.5% on average on GLUE. We publish the selected dataset and the code for DSIR to improve future LM pretraining. [1]

## 2 Setup

Given a small number of target text examples $x'_1, x'_2, ..., x'_n$ from a target distribution of interest $p$ and a large raw dataset $x_1, x_2, ..., x_N$ drawn from distribution $q$, we aim to select $k$ examples ($k \ll N$) from the raw dataset that are similar to the target.

**Selection via heuristic classification.**    As a starting point, we first define the heuristic classification method used by GPT-3/The Pile/PaLM [10; 21; 14]. In heuristic classification, we train a binary classifier $f : \mathcal{X} \to [0,1]$ to output the probability that an input is sampled from the target distribution. The model is typically a fasttext linear classifier on n-gram feature vectors (usually unigrams and bigrams) [30]. We initialize the feature vectors from pretrained fasttext word vectors. We use the trained classifier to estimate $f(x_i)$, the predicted probability that $x_i$ is sampled from the target, for all raw examples. Example $x_i$ is selected if $f(x_i) > 1 - \beta_i$, where $\beta_i$ is a sample from a Pareto distribution (typically with shape parameter $\alpha = 9$ [10; 14]). Since each example is kept or discarded independently, to select a desired number of examples $k$, the process must either be repeated or $\alpha$ must be tuned. Heuristic classification selects examples from modes of the target distribution (high $f(x_i)$), which could lack diversity. To combat this, noise $\beta_i$ is added. However, it is unclear how much noise to add and there are no guarantees on the selected data distribution.

## 3 Data Selection with Importance Resampling

In the DSIR framework, we consider using importance resampling [68] to select examples that are distributed like the target. However, estimating importance weights on high dimensional data like text is often statistically intractable without sufficient additional structure [6; 75; 23].

Instead, we employ importance resampling on a feature space $\mathcal{Z}$ that provides this structure. DSIR uses a feature extractor $h : \mathcal{X} \to \mathcal{Z}$ to map the input $x$ to features $z = h(x)$. The induced raw and target feature distributions are $q_{\text{feat}}$ and $p_{\text{feat}}$, respectively. The goal is to select examples with features that are approximately distributed according to the target feature distribution $p_{\text{feat}}$. Depending on the choice of feature extractor, DSIR focuses on different aspects of the input. For example, an n-gram feature extractor focuses on matching the n-gram frequencies of the selected data and the target.

Our framework consists of 3 steps:

1. *Learn $\hat{p}_{feat}$ and $\hat{q}_{feat}$:* We learn two feature distributions $\hat{p}_{\text{feat}}$ and $\hat{q}_{\text{feat}}$ using held-out featurized examples from the target and raw data, respectively.

2. *Compute importance weights:* We compute the importance weights $w_i = \frac{\hat{p}_{\text{feat}}(z_i)}{\hat{q}_{\text{feat}}(z_i)}$ for each featurized example $z_i = h(x_i)$ from the $N$ raw examples.

3. *Resample:* Sample $k$ examples without replacement from a categorical distribution with probabilities $\frac{w_i}{\sum_{i=1}^N w_i}$. Sampling without replacement avoids choosing the same example multiple times, is more statistically efficient for importance resampling [23], and can be implemented efficiently with the Gumbel top-$k$ trick [81; 38; 87; 39].

## 4 DSIR with Hashed N-gram Features

For efficiency and scalability, we instantiate DSIR with hashed n-gram features. Later in Section 6, we test whether hashed n-gram features preserve the information needed to select data relevant to the target.

**Hashed n-gram features.**    We consider hashed n-gram features inspired from fasttext [30; 86]. Specifically, for each example $x$ we form a list of unigrams and bigrams, hash each n-grams into one of $m$ buckets ($m = 10000$ in this paper), and return the counts of the hashed buckets in a $m$-dimensional feature vector $z \in \mathbb{N}^m$. For example, if the text input is "Alice is eating", we form the list [Alice, is, eating, Alice is, is eating], hash each element of this list to get a list of indices [1, 3, 3, 2, 0] and return the vector of counts for each index [1, 1, 1, 2, ... ]. While the hashing

introduces some noise due to collisions, we find that this is a simple and effective way to incorporate both unigram and bigram information.

**Bag of hashed n-grams model.** We parameterize the raw and target feature distributions $p_{\text{feat}}$ and $q_{\text{feat}}$ as bag-of-ngrams models. The bag-of-ngrams model has parameters $\gamma \in \Delta^m$, which is a vector of probabilities on the hash buckets that sums to 1, Under this model, the probability of a feature vector $z \in \mathbb{N}^m$ is

$$\mathbb{P}(z;\gamma) = \prod_{j=1}^{m} \gamma[j]^{z[j]} \tag{1}$$

where the bracket notation selects the corresponding index in the vector. Given some featurized examples $\tilde{z}_1, \ldots, \tilde{z}_s$ sampled from a feature distribution, we estimate the parameters by counting: $\hat{\gamma} = \frac{1}{\sum_{i=1}^{s} \mathbf{1}^\top \tilde{z}_i} \sum_{j=1}^{s} \tilde{z}_j$.

**Speed benchmark on The Pile.** To test the scalability of the framework, we benchmark DSIR with hashed n-gram features on selecting data from the full Pile dataset [21]. For this test, we do not preprocess the data other than decompressing the text files for faster I/O. We use hashed n-gram features with 10k buckets, fit the raw feature distribution with 1B hashed indices from the Pile, and fit the target feature distribution with the full target dataset (ChemProt [41]). DSIR selects 100M documents from the full Pile dataset in 4.5 hours on 1 CPU node with 96 cores. Almost all of the time (4.36 hours) is spent computing the importance weights on the raw dataset, while fitting the feature distributions (1 minutes) and resampling (6 minutes) were much faster. Increasing the number of CPU cores can further decrease the runtime.

# 5 Selecting Data for Domain-Specific Continued Pretraining

In this section, we use DSIR to select domain-specific data for continued pretraining. We compare DSIR to 7 other data selection methods in this continued pretraining setting.

**Setup.** We select data for 8 target distributions in the setting of Gururangan et al. [24], where we perform continued pretraining of domain-specific LMs. Here, the target is a specific downstream unlabeled data distribution and we select examples from The Pile (the raw data). For each downstream dataset, we select data for continued pretraining starting from RoBERTa [48] (see Appendix H). Following Gururangan et al. [24], we consider 8 downstream datasets across 4 domains: Computer Science papers (ACL-ARC [31], Sci-ERC [51]), Biomedicine (ChemProt [41], RCT [16]) News (AGNews [92], HyperPartisan [34]), and Reviews (Helpfulness [53], IMDB [52]).

**Baselines.** Beyond random selection (without replacement) and heuristic classification, we also compare against manual curation [24] and a top-$k$ variant of heuristic classification. In manual curation, we simply fine-tune from domain-adaptive pretraining (DAPT) checkpoints [24], which are the result of continued pretraining on manually-curated data. In top-$k$ heuristic classification, we select the top-$k$-scoring examples according to the binary classifier used in heuristic classification. All methods select data from The Pile except for manual curation, which uses domain-specific data sources [24].

We perform a controlled comparison by equalizing the amount of LM training compute for all methods, measured by the number of tokens processed during training, following the compute budget in Gururangan et al. [24]. For random selection, heuristic classification, and DSIR using n-gram features (defined in Section 4), we control the number of selected examples (25M examples with fixed token length 256) and the training protocol. We standardize the fine-tuning for all models and average all results over 5 random seeds (see Appendix H for details). All the models initialize from RoBERTa-base. Before data selection via DSIR or heuristic classification, we remove extremely short (<40 words) or repetitive documents that tend to be uninformative (Appendix J).

**Automatic data selection with DSIR can replace manual curation.** Table 1 shows the comparison between the data selection methods. To summarize:

- On average, DSIR improves over random selection by 1.2% and manually curated data (DAPT) by 0.3%, showing the potential to replace manual curation.

Table 1: F1 scores for continued pretraining from the RoBERTa checkpoint [48] on 8 downstream datasets from 4 domains (CS, Biomed, News, and Reviews). Random selection, heuristic classification, and DSIR train on 25M selected examples from The Pile. Heuristic classification and DSIR create a different pretraining dataset for every downstream dataset. All models (including DAPT [24]) use the same amount of training compute and results are averaged over 5 seeds, with standard deviations in subscripts. All datasets use macro-F1 except ChemProt and RCT, which use micro-F1.

| | ACL-ARC | Sci-ERC | ChemProt | RCT | HyperPartisan | AGNews | Helpfulness | IMDB | Avg |
|---|---|---|---|---|---|---|---|---|---|
| RoBERTa (no continued pretrain) | $66.80_{1.08}$ | $80.14_{2.25}$ | $82.31_{0.54}$ | $86.68_{0.14}$ | $\mathbf{88.85}_{2.59}$ | $93.35_{0.2}$ | $65.08_{2.29}$ | $94.38_{0.13}$ | 82.20 |
| Random selection | $67.51_{2.60}$ | $\mathbf{80.53}_{1.65}$ | $83.14_{0.52}$ | $86.85_{0.13}$ | $86.42_{5.33}$ | $93.52_{0.15}$ | $68.15_{1.37}$ | $94.49_{0.25}$ | 82.58 |
| Manual curation/DAPT [24] | $71.84_{4.78}$ | $80.42_{1.57}$ | $84.17_{0.50}$ | $87.11_{0.10}$ | $87.23_{3.65}$ | $93.61_{0.12}$ | $68.21_{1.07}$ | $\mathbf{95.08}_{0.11}$ | 83.46 |
| Heuristic classification | $69.94_{2.96}$ | $80.52_{0.95}$ | $83.35_{1.07}$ | $86.78_{0.17}$ | $85.71_{6.01}$ | $93.54_{0.19}$ | $68.50_{0.79}$ | $94.66_{0.22}$ | 82.88 |
| Top-$k$ Heuristic classification | $71.73_{0.21}$ | $80.22_{0.58}$ | $84.11_{0.73}$ | $87.08_{0.21}$ | $88.29_{8.28}$ | $\mathbf{93.67}_{0.14}$ | $\mathbf{69.18}_{0.73}$ | $94.90_{0.14}$ | 83.65 |
| DSIR | $\mathbf{72.86}_{2.71}$ | $80.44_{1.13}$ | $\mathbf{85.51}_{0.46}$ | $\mathbf{87.14}_{0.13}$ | $87.01_{4.53}$ | $93.62_{0.19}$ | $68.95_{0.81}$ | $94.56_{0.34}$ | $\mathbf{83.76}$ |

Table 2: F1 scores for continued pretraining from RoBERTa, testing different components of heuristic classification and DSIR methods. For heuristic classification, replacing the Pareto noisy threshold with calibration and importance resampling (DSIR (fasttext discriminative)) improves F1. Generative importance weight estimators outperform discriminative importance weight estimators for DSIR. All results average over 5 seeds, with standard deviations in subscripts.

| | ACL-ARC | Sci-ERC | ChemProt | RCT | HyperPartisan | AGNews | Helpfulness | IMDB | Avg |
|---|---|---|---|---|---|---|---|---|---|
| Heuristic classification | $\mathbf{69.94}_{2.96}$ | $\mathbf{80.52}_{0.95}$ | $83.35_{1.07}$ | $86.78_{0.17}$ | $85.71_{6.01}$ | $\mathbf{93.54}_{0.19}$ | $\mathbf{68.50}_{0.79}$ | $94.66_{0.22}$ | 82.88 |
| DSIR (n-gram generative) | $\mathbf{72.86}_{2.71}$ | $80.44_{1.13}$ | $\mathbf{85.51}_{0.46}$ | $\mathbf{87.14}_{0.13}$ | $87.01_{4.53}$ | $93.62_{0.19}$ | $\mathbf{68.95}_{0.81}$ | $94.56_{0.34}$ | $\mathbf{83.76}$ |
| DSIR (fasttext discriminative) | $68.46_{7.15}$ | $79.00_{1.50}$ | $\mathbf{84.57}_{0.65}$ | $87.09_{0.08}$ | $\mathbf{89.18}_{4.06}$ | $93.54_{0.14}$ | $68.41_{1.51}$ | $\mathbf{94.95}_{0.29}$ | 83.15 |
| DSIR (n-gram discriminative) | $70.35_{2.90}$ | $80.21_{0.85}$ | $85.03_{1.18}$ | $87.04_{0.19}$ | $85.49_{8.24}$ | $\mathbf{93.74}_{0.07}$ | $68.79_{1.22}$ | $94.84_{0.24}$ | 83.19 |
| DSIR (unigram generative) | $69.53_{0.16}$ | $79.69_{1.91}$ | $85.24_{0.88}$ | $87.05_{0.10}$ | $\mathbf{90.11}_{5.39}$ | $93.42_{0.16}$ | $68.55_{0.78}$ | $94.39_{0.33}$ | 83.50 |

- DSIR improves over heuristic classification by 0.9% and is comparable to top-$k$ heuristic classification. We note that top-$k$ heuristic classification is not typically used in this setting, but we find that it may be particularly suited for domain-specific data selection, where diversity may be less important than the general-domain setting.

- Random selection improves by 0.4% on average over no continued pretraining at all, showing that additional data generally improves the downstream F1 score. All the targeted data selection methods improve over random selection.

**Discriminative importance weight estimators underperform generative estimators.** We experiment with replacing components of DSIR in Table 2. First, we consider using the binary classifier from heuristic classification (which takes pretrained fasttext word vectors as input) as the importance weight estimator in DSIR. For input $x_i$, the classifier predicts the probability of target $f(x_i)$. We use this to estimate importance weights $\frac{f(x_i)}{1-f(x_i)}$, then resample according to these weights. This approach (DSIR (fasttext discriminative)) improves F1 by 0.3% over heuristic classification. However, this approach still underperforms DSIR by 0.6% on average, even with regularization and calibration.

We consider another discriminative version of DSIR that uses hashed n-gram features as input to a logistic regression binary classifier for importance weight estimation. This differs from heuristic classification, which initializes with pretrained fasttext feature vectors and fine-tunes the features along with the classifer. This approach (DSIR (n-gram discriminative)) underperforms DSIR by 0.7%, even with regularization and calibration. These results suggest that a generative approach is better suited (or easier to tune) for importance resampling. However, the discriminative approaches still outperform random selection by 0.6%.

**Selecting with n-grams improves downstream performance over unigrams.** DSIR uses both unigram and bigram information to compute hashed n-gram features. We ablate the role of bigram information in hashed n-grams by using hashed unigram features (with 10k buckets) for DSIR. In Table 2, we find that DSIR with unigram features underperforms DSIR with n-grams by 0.26%, though still achieving comparable F1 score to manual curation. Overall, selecting data with unigrams is effective, but including bigrams further improves the relevance of the selected data.

**Cross-domain analysis and the effect of the choice of pretraining data.** DSIR assumes knowledge of the target distribution, but what happens if the target dataset is not representative of the target

Figure 2: F1 scores of DSIR for all pairs of pretraining data target distributions (rows) and downstream tasks (columns). The cells are colored by its per-column ranking, with better rankings (higher F1 scores) having darker colors. While using the pretraining data selected specifically for the downstream task is typically strong, choosing the worst pretraining dataset for the downstream task reduces F1 by 6% on average. All results are averaged over 5 seeds.

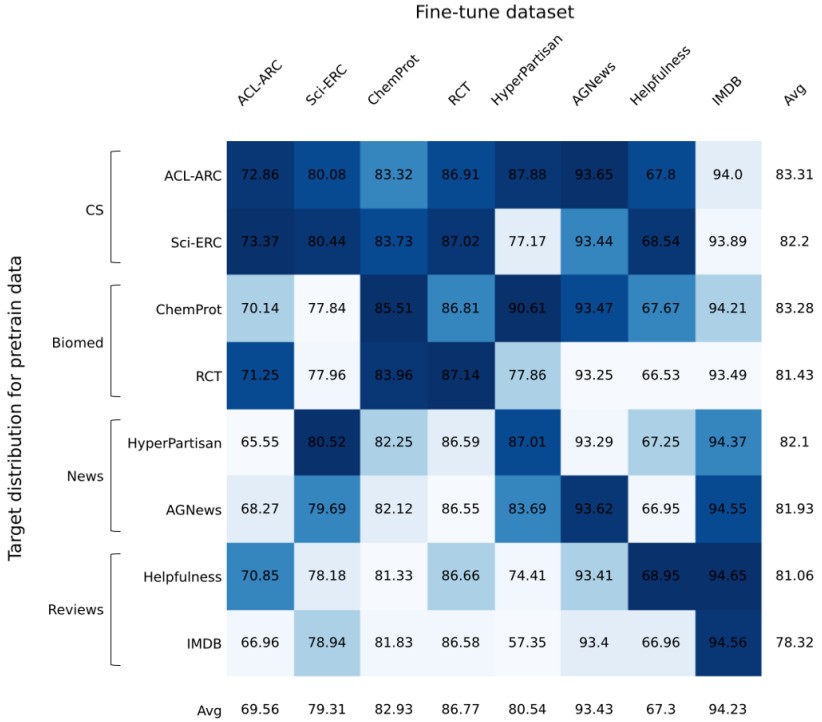

distribution? To test the effect of varying the target distribution on downstream performance, we consider every pair of pretraining dataset, which is selected by DSIR for target downstream task X, and downstream task Y. Figure 2 provides the full matrix of results. We find a 6% average drop in F1 when we choose the worst pairing for each downstream task instead of matching the pretraining and downstream data. In the worst case, the F1-score on HyperPartisan drops by 30%. Thus, the choice of target distribution can have a large effect on downstream performance.

**Pretraining data transfers better for targets within the same domain.** In practice, we may have access to some target datasets in the relevant domain and hope to select pretraining data that can improve performance on other tasks in that domain. The 8 target/downstream tasks we use come from 4 domains, with 2 tasks from each domain. We define within-domain F1 as the average F1 of the pairs of pretraining and fine-tuning data from the same domain, but excluding pairs where the pretraining data is selected for the fine-tuning task. We compute this by averaging the off-diagonal elements in the $2 \times 2$ diagonal blocks of the matrix in Figure 2. We find that the within-domain F1 (82.9%) is 1.7% higher on average than the cross-domain F1 (81.2%), where the pretraining data is selected for a target from a different domain.

## 6 KL Reduction on Hashed N-grams Predicts Downstream Performance

When designing a feature extractor for DSIR, how do we measure whether the features preserve the information for selecting relevant pretraining data? To answer this question, we propose a data metric, *KL reduction*, which measures how much data selection reduces distance to the target over random selection in a feature space. We find that KL reduction on hashed n-gram features strongly correlates with downstream performance across various data selection methods, including those that do not involve n-grams, such as manual curation.

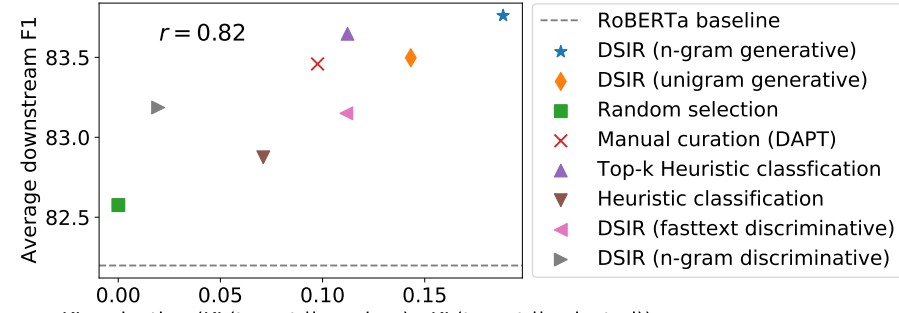

Figure 3: Plot of average KL reduction on the n-gram feature space, defined as how much the selected dataset reduces KL divergence to the target distribution over just random sampling from The Pile, against average downstream F1 score over the 8 continued pretraining datasets in Table 1. There is a strong correlation between KL reduction and downstream performance (Pearson $r = 0.82$).

**KL reduction metric.** We define *KL reduction* as the average reduction in empirical KL divergence from doing data selection over random selection over a set of target feature distributions $\mathcal{T}$:

$$\text{KL-reduction}(p'_{\text{feat}}; \hat{q}_{\text{feat}}, \mathcal{T}) = \frac{1}{|\mathcal{T}|} \sum_{\hat{p}_{\text{feat}} \in \mathcal{T}} \text{KL}(\hat{p}_{\text{feat}} \| \hat{q}_{\text{feat}}) - \text{KL}(\hat{p}_{\text{feat}} \| p'_{\text{feat}}) \tag{2}$$

where $p'_{\text{feat}}$ is the empirical feature distribution of the selected data, $\hat{p}_{\text{feat}}$ is a empirical target feature distribution, $\hat{q}_{\text{feat}}$ is the empirical raw feature distribution. KL reduction depends on the raw distribution $\hat{q}_{\text{feat}}$ and the set of target distributions $\mathcal{T}$ as hyperparameters. In our continued pretraining setting, $\hat{q}_{\text{feat}}$ is the feature distribution of the Pile and $\mathcal{T}$ consists of the feature distributions from the 8 downstream tasks from Section 5.

**KL reduction on hashed n-grams predicts downstream performance.** We show that when computed on the hashed n-gram feature space, KL reduction of a selected dataset highly correlates with the downstream performance of a model trained on that data. Figure 3 plots KL reduction against average downstream performance over 8 target distributions for 8 data selection methods from The Pile [21], where the distribution parameters are estimated using 100k samples from each dataset. The average downstream F1 score is highly correlated with the KL reduction (Pearson $r = 0.82$). This agrees with the results of Razeghi et al. [65] for in-context learning [10] and extends the preliminary evidence from Gururangan et al. [24] on one selection method that better unigram overlap improves downstream performance. DSIR with hashed n-gram features achieves the highest KL reduction and the best average downstream F1.

While some of the original pretraining datasets for DAPT [24] were not publicly available, we downloaded the public versions as an approximation. Our results suggest that hashed n-gram features preserve most of the information needed for selecting data relevant to the target. Since KL reduction highly correlates with downstream performance and can be cheaply computed without training an LM, KL reduction can be used as a sanity check for future data selection methods.

## 7 Selecting Data for Training General-Domain LMs

In this section, we consider selecting formal text (as a proxy for high-quality text) for training general-domain LMs from scratch. We use Wikipedia and books as the target distribution.

**Baselines and setup.** We compare the following methods for selecting data from the Pile: 1) Random selection, 2) Heuristic classification (GPT-3/Pile/PaLM method), and 3) DSIR. As ablations, we consider top-$k$ variants of heuristic classification and DSIR (take the top-$k$ examples according to importance weights instead of resampling). We use each method to select 51.2M examples, which corresponds to 4 epochs with our compute budget. For heuristic classification and DSIR, we select 96% of the examples from domains excluding Wikipedia and books. This is done to reduce the bias towards selecting data from Wikipedia and books (the target distribution). We choose the other 4%

Table 3: Accuracies on the GLUE [82] dev set for a BERT-style masked language model [17] trained on data selected from The Pile [21]. Following RoBERTa [48], for RTE, STS, and MRPC we fine-tune starting from the MNLI model instead of from scratch. DSIR outperforms heuristic classification (used by GPT-3 and PaLM) and random selection by over 2% on average. All results are averaged over 5 seeds and standard deviations are in subscripts.

| | MNLI | QNLI | QQP | RTE | SST-2 | MRPC | CoLA | STS-B | Avg |
|---|---|---|---|---|---|---|---|---|---|
| Random selection | $82.63_{0.41}$ | $86.90_{0.28}$ | $89.57_{0.30}$ | $67.37_{1.69}$ | $90.05_{0.41}$ | $87.40_{1.08}$ | $49.41_{3.67}$ | $88.63_{0.11}$ | 80.25 |
| Heuristic classification | $82.69_{0.17}$ | $85.95_{0.79}$ | $89.77_{0.32}$ | $68.59_{1.75}$ | $88.94_{0.98}$ | $86.03_{0.93}$ | $48.17_{3.19}$ | $88.62_{0.22}$ | 79.85 |
| Top-$k$ Heuristic classfication | $83.34_{0.22}$ | $88.62_{0.24}$ | $89.89_{0.19}$ | $70.04_{0.99}$ | $91.15_{0.76}$ | $86.37_{1.00}$ | $53.02_{3.56}$ | $89.30_{0.11}$ | 81.47 |
| DSIR | $83.07_{0.29}$ | $\mathbf{89.11}_{0.14}$ | $89.80_{0.37}$ | $\mathbf{75.09}_{2.76}$ | $90.48_{0.57}$ | $\mathbf{87.70}_{0.68}$ | $\mathbf{54.00}_{1.34}$ | $89.17_{0.13}$ | $\mathbf{82.30}$ |
| Top-$k$ DSIR | $\mathbf{83.39}_{0.06}$ | $88.63_{0.38}$ | $\mathbf{89.94}_{0.17}$ | $72.49_{1.29}$ | $\mathbf{91.01}_{0.79}$ | $86.18_{1.12}$ | $49.90_{1.10}$ | $\mathbf{89.52}_{0.21}$ | 81.38 |

uniformly from Wikipedia and books, and did not tune these proportions (Appendix F). We apply a quality filter for extremely short or repetitive examples before heuristic classification and DSIR selection (Appendix J). For each dataset, we perform MLM pretraining for 50k steps with a large batch size (4096) and short token length (128), following Izsak et al. [28]. All the models use the BERT-base architecture [17]. We evaluate the models on the GLUE dev set, averaged over 5 fine-tuning runs [82]. Fine-tuning hyperparameters such as the number of epochs and batch size are fixed for each dataset, following reasonable defaults from the RoBERTa codebase [48].

**DSIR qualitatively selects more formal text.** Figure 4 shows the beginning characters of 20 random examples selected by random selection, heuristic classification, and DSIR. The random sample contains many code examples that are not similar to text from Wikipedia and books. Heuristic classification seems slightly too diverse, which suggests that the variance of the Pareto distribution added to the classifier scores may be too high. Note that we use the setting of the Pareto shape hyperparameter used in GPT-3 [10]. Qualitatively, DSIR selects the most formal text. By doing importance resampling to match the target distribution, DSIR trades off the relevance and diversity of the selected data automatically.

**DSIR improves GLUE performance.** Table 3 shows results on the GLUE dev set. DSIR achieves 82.3% average GLUE accuracy, improving over random selection by 2% and heuristic classification by 2.5%. Heuristic classification leads to 0.5% lower accuracy than random selection from The Pile. We hypothesize this is because The Pile has already been filtered once with heuristic classification.

**Resampling outperforms top-$k$ selection.** Top-$k$ heuristic classification and top-$k$ DSIR have similar performance across datasets, with some tradeoffs compared to DSIR without top-$k$. DSIR without top-$k$ is competitive with these variants on all datasets and achieves a 0.8–0.9% higher average over top-$k$ variants. All the top accuracies across datasets are achieved by DSIR or top-$k$ DSIR.

| (a) Random selection | (b) Heuristic classification | (c) DSIR |
|---|---|---|
| alignment. The value of Z~k | in London and, like | when ship rats invaded. |
| simultaneously. To prove | raid5 sf if by accident | Stephon Gilmore\nBuffalo |
| $p^{g}_{\phi}(x)=$ | denied that the plaintiff | This story is a sequel |
| \\; r \\; e^{−\alpha z} | Are celebrities the new | BLM to Begin Massive |
| suggests that, nicotinic | made on the antimesenteric | Below, in alphabetical |
| THE AUTHORS OR COPYRIGHT | woodland species consumed | enforcement, the office |
| H&E coupe over a length | they differ from gecko | O'Brien is a Senior |
| if (codingState == SMModel | that someone may legit | Lockwood, commander of |
| Five lessons in viral | console.log(err); | A ten−year−old girl |
| {m}_s}{{m}_b}\\,.$$ | \\usepackage{wasysym} | the suburbs, a regular |
| Omega, which automatically | ∗.∗",,,, True) | American lawyers\nCategory: |
| C1−−−O1−−−Cu2 128.2\xa0(2) | Prison officials deposed | state's chief executive. |
| if the given properties | plan to see it 30 more." | of a scheduled program, |
| ∗ GPL Classpath Exception: | Misdiagnosis of mosaic | Guard unit was identifiably |
| husband and I quit donating | haven\'t had issues when | have to put up the word |
| The helical character of | slowly over a napkin. | Indiana head coach Archie |
| new TestWriteLine(" | In turn, this may trigger | a committee comprised of |
| 6−month period in 2014−−15 | 50\u2009ul of 20\u2009mM | Filipacchi Media said |
| "Alice Adams: An Intimate | we are afraid of patrons. | Anna had the necessary |
| index : m_outerStart); | Linas Gylys, president | a few days ago. |

Figure 4: Beginning characters of 20 random examples (each line is a different example) selected by random selection, heuristic classification, and DSIR, where the target is formal text from Wikipedia + books. Qualitatively, DSIR selects more formal text than random selection and heuristic classification.

# 8  Related Work

**Effect of pretraining data on LMs.**   The pretraining data has a large effect on LM performance. Lee et al. [44]; Hernandez et al. [26] show that deduplicating data improves LMs, and Baevski et al. [3]; Yang et al. [88] compare using a large web corpus versus Wikipedia. Raffel et al. [64] shows that heuristically filtered data (filtering out short and duplicated examples) improves T5 and  Du et al. [18] shows that heuristic classification improves downstream few-shot performance for GLaM. We provide extensive controlled experiments comparing the effect of data selection methods on downstream performance.

**Retrieval.**   Yao et al. [89] use keyword-based retrieval (BM25) to select data for semi-supervised learning.  In preliminary tests, we found that out of 6.1M documents retrieved by BM25, there were only 1.8M unique documents (70% were exact duplicates). These duplicate examples can hurt performance [44; 26]. Selecting a desired number of unique documents involves oversampling and de-duplication.  Instead, we consider top-$k$ heuristic classification, which has similarities to cosine similarity-based retrieval (since heuristic classification uses an inner product score between pretrained word embeddings and a learned class vector) and avoids retrieving repeated examples.

**Data selection in classical NLP.**   Moore-Lewis selection [56; 2; 20] takes the top-$k$ examples in cross-entropy difference between n-gram LMs trained on target and raw data to score examples, which could over-sample examples from the mode of the target distribution. In Section 7, we found that top-$k$ DSIR, which is a form of Moore-Lewis selection with hashed n-gram LMs, underperforms DSIR by 0.9% on GLUE. DSIR naturally balances diversity and relevance for use in both domain-specific and general-domain cases, since it uses importance resampling to match the target distribution. Feature-space/n-gram discrepancy measures [29; 69; 47] have also been used in selecting data in the domain adaptation setting.  Overall, these methods do not consider importance resampling and do not address the gap between pretraining and downstream tasks: pretraining has a different objective to fine-tuning, pretraining uses unlabeled data that is not task-formatted, and the influence of pretraining data is separated from the final model by the fine-tuning step. Beyond the preliminary evidence that unigram similarity metrics are related to downstream performance in Gururangan et al. [24], we show comprehensively and quantitatively on 8 selection methods that despite the pretrain-downstream gap, n-gram KL reduction on pretraining datasets highly correlates with downstream performance.

**Data selection in deep learning.**   Many works show the importance of data selection in the supervised or semi-supervised learning setting in vision [76; 54; 33; 36; 35; 37; 83; 85; 61; 55; 15; 71; 5] and in language finetuning [15; 54]. While most select image data from CIFAR or ImageNet, which have up to 1–10M examples, we consider selecting text data from The Pile, which has over 1.6B examples (of 128 whitespace-delimited words each). At this scale, previous methods become quite expensive since they typically require running a neural network forward pass to get embeddings [71; 76; 37], taking gradients [35; 36; 83; 61; 55], or training a reference model [54]. In contrast, we construct a simple n-gram-based selection method that easily scales to internet-scale datasets.  Coleman et al. [15] select data with high uncertainty under a smaller proxy neural model. They do not consider using a target dataset for estimating importance weights. However, using a neural model could be a complementary strategy for importance resampling. Other works [70; 50; 32] focus on choosing a subset that approximates training with the original dataset and require selecting data online during training. We aim to select a targeted dataset (once, before training) with different properties from the raw data (restricting the data to formal text or a specific domain). Our work also differs from active learning methods [72; 19; 84; 90; 80], which query an annotator for more labeled data. Instead, we select data for self-supervised pretraining.

**Importance weighting and domain adaptation.**   Many methods tackle the high-dimensional importance weight estimation problem [79; 67; 12; 13]. In particular, importance weighting is classically used in domain adaptation [74; 78], where unlabeled target examples are used to adapt a model trained on labeled source data, for reweighting the loss function. However, in many modern applications the source and target are often disjoint (e.g., sketches vs. natural images), causing undefined importance weights [62; 42; 73]. We side-step high-dimensional importance weight estimation by instead working in a reduced feature space where the support of the massive web corpus should cover the target.

# 9 Discussion and Limitations

**Feature space for importance resampling.** Finding an appropriate feature space is important for DSIR. Although we find a tight correlation between downstream performance and our data metric compute using hashed n-gram features, n-grams only capture a superficial word-level overlap. Other feature extractors, such as neural models, may produce features that better capture semantics. We consider a variant of DSIR which estimates importance weights on a neural feature space in Appendix B, and find that this variant also improves by 1–1.5% over random selection and heuristic classification on GLUE, but our preliminary version does not improve over DSIR with hashed n-gram features. However, extracting these features is much more computationally expensive (on the order of $D$ times more FLOPs for a $D$-parameter neural model), and importance weight estimation on this continuous feature space may be more difficult.

**Parameterization of the importance weight estimator.** In principle, both generative and discriminative approaches to estimating the importance weights should work. In a discriminative approach, regularization and calibration should be used to combat overfitting and make the predicted probabilities useful for importance resampling. We find that a generative approach requires less tuning and could also be better when the number of target examples is small, as Ng and Jordan [58] finds that Naive Bayes often performs better than logistic regression in low-sample regimes.

**What is the right target distribution?** When developing a domain-specific model such as Codex [11], the target dataset should be representative of the coding tasks we expect the model to be used on. However, it's unclear how exactly to collect this dataset and how much to weight each task in the target distribution. Developing better procedures for collecting the target dataset can ultimately improve the data selected by DSIR. For general-domain LMs, we follow GPT-3, the Pile, and PaLM in using formal text from Wikipedia and books as a proxy for high quality text [10; 21; 18; 14]. However, this is just a heuristic. We leave the exploration of other target distributions for general-domain LMs to future work.

**Broader impacts.** The impact of DSIR depends on the properties of the target data. While DSIR could amplify biases present in the target examples, with the appropriate target data, DSIR can be used to collect data that improve the training efficiency, alignment, or bias of LMs [59; 4; 40]. These benefits could reduce the environmental impact of LMs [77; 43; 46; 60] and reduce their biases and risks [9; 1; 22; 8]. For example, DSIR can be used to collect more data on underrepresented subpopulations and fine-tune the model on this data to improve model fairness.

# 10 Conclusion

We provide a cheap and scalable data selection framework based on importance resampling for improving the downstream performance of LMs. We also find a data metric, KL reduction, that strongly correlates with downstream performance and can provide a sanity check for data selection methods without training a model. Our work provides a step in understanding the choice of pretraining data for downstream transfer in LMs.

# 11 Acknowledgements

We thank Neil Band, Hong Liu, Garrett Thomas, and anonymous reviewers for their feedback. This work was supported by an Open Philanthropy Project Award and NSF IIS 2211780. SMX was supported by a NDSEG Fellowship. SS is supported by an Open Philanthropy Graduate Fellowship.

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

# A  DSIR asymptotically selects from the target

We prove that DSIR selects examples with features distributed as the target as the raw dataset size goes to infinity, assuming that the importance weights are correct up to a constant factor.

**Proposition A.1.** *Assume that the importance weights $w_i$ are proportional to the true importance weights $\frac{p_{feat}(z_i)}{q_{feat}(z_i)}$. Then as the number of raw examples $N$ goes to infinity, the procedure returns $k$ i.i.d. samples with features distributed according to the target feature distribution $p_{feat}$.*

*Proof.* By assumption, we have importance weights $w_i$ that are proportional to the true importance weights, so that $w_i = C\frac{p_{feat}(z_i)}{q_{feat}(z_i)}$ for the $i$-th source example for some constant $C > 0$. First suppose that $k = 1$. Then,

$$\text{Prob. of sampling an example with feature value } z = \frac{\sum_{i=1}^{N}\mathbf{1}[z_i = z]w_i}{\sum_{j=1}^{N}w_j} \tag{3}$$

$$= \frac{C\sum_{i=1}^{N}\mathbf{1}[z_i = z]\frac{p_{feat}(z_i)}{q_{feat}(z_i)}}{\sum_{j=1}^{N}C\frac{p_{feat}(z_j)}{q_{feat}(z_j)}} \tag{4}$$

$$= \frac{\frac{1}{N}\sum_{i=1}^{N}\mathbf{1}[z_i = z]\frac{p_{feat}(z_i)}{q_{feat}(z_i)}}{\frac{1}{N}\sum_{j=1}^{N}\frac{p_{feat}(z_j)}{q_{feat}(z_j)}}. \tag{5}$$

For $k \geq 1$, we can similarly compute the probability of sampling the $m$-th example ($m \in \{1,...,k\}$) as:

$$\text{Prob. of sampling } m\text{-th example with feature value } z = \frac{\frac{1}{N-m+1}\sum_{i=1}^{N-m+1}\mathbf{1}[z_i = z]\frac{p_{feat}(z_i)}{q_{feat}(z_i)}}{\frac{1}{N-m+1}\sum_{j=1}^{N-m+1}\frac{p_{feat}(z_j)}{q_{feat}(z_j)}}, \tag{6}$$

where for notational convenience, we re-index the raw examples after selecting each example.

For each $m \in \{1,...,k\}$, the numerator converges to $p_{feat}(z)$ as $N \to \infty$:

$$\frac{1}{N-m+1}\sum_{i=1}^{N-m+1}\mathbf{1}[z_i = z]\frac{p_{feat}(z_i)}{q_{feat}(z_i)} = \frac{1}{N-m+1}\sum_{i=1}^{N-m+1}\mathbf{1}[z_i = z]\frac{p_{feat}(z)}{q_{feat}(z)} \to q_{feat}(z)\frac{p_{feat}(z)}{q_{feat}(z)} = p_{feat}(z) \tag{7}$$

since $z_j$ (raw features) are sampled from $q_{feat}$ (raw feature distribution). For the same reason, the denominator converges to 1:

$$\frac{1}{N-m+1}\sum_{j=1}^{N-m+1}\frac{p_{feat}(z_j)}{q_{feat}(z_j)} \to \mathbb{E}_{q_{feat}}\left[\frac{p_{feat}(z_j)}{q_{feat}(z_j)}\right] = 1. \tag{8}$$

Therefore the features of the $m$-th example is sampled from $p_{feat}$ for all $m \in \{1,...,k\}$. $\qquad\square$

**Intuition from a simple example.**  DSIR uses importance resampling to better balance the tradeoff between relevance and diversity as the samples converge to true samples from the target distribution (Proposition A.1), while no such guarantee holds for top-k selection. For intuition using a simple example, consider a raw dataset of $n$ coin flips from a biased coin, with $0.9n$ heads and $0.1n$ tails. We want to filter the raw dataset to have $k = 10$ flips from a fair coin (the target distribution). The importance weights are $\frac{1}{2\cdot0.9}$ for the heads examples and $\frac{1}{2\cdot0.1}$ for the tails examples (the tails have higher weight). If we select the top $k$ flips according to the importance weight, we will select 10 tails, still resulting in a biased dataset. However, importance resampling balances this out, resulting in a fair dataset in expectation as $n$ goes to infinity. We ran a simulation of the simple example with $k = 10$ and varying raw data sizes $n$ to see how fast the resampled dataset converges to a fair dataset with the raw data size. For raw data sizes $n \in \{100,200,500\}$, DSIR selects a dataset with (44%, 47%, 50%) heads respectively, averaged over 1000 trials. Thus, DSIR converges quickly to the desired target distribution. In all cases, top-$k$ selects a dataset with all tails.

Table 4: Accuracies on the GLUE [82] dev set for a BERT-style masked language model [17] trained on data selected from The Pile [21]. Following RoBERTa [48], for RTE, STS, and MRPC we fine-tune starting from the MNLI model instead of from scratch. DSIR outperforms heuristic classification (used by GPT-3 and PaLM) and random selection by over 2% on average. All results are averaged over 5 seeds and standard deviations are in subscripts.

| | MNLI | QNLI | QQP | RTE | SST-2 | MRPC | CoLA | STS-B | Avg |
|---|---|---|---|---|---|---|---|---|---|
| Random selection | $82.63_{0.41}$ | $86.90_{0.28}$ | $89.57_{0.30}$ | $67.37_{1.69}$ | $90.05_{0.41}$ | $87.40_{1.08}$ | $49.41_{3.67}$ | $88.63_{0.11}$ | 80.25 |
| Heuristic classification | $82.69_{0.17}$ | $85.95_{0.79}$ | $89.77_{0.32}$ | $68.59_{1.75}$ | $88.94_{0.98}$ | $86.03_{0.93}$ | $48.17_{3.19}$ | $88.62_{0.22}$ | 79.85 |
| Top-$k$ Heuristic classfication | $83.34_{0.22}$ | $88.62_{0.24}$ | $89.89_{0.19}$ | $70.04_{0.99}$ | $91.15_{0.76}$ | $86.37_{1.00}$ | $53.02_{3.56}$ | $89.30_{0.11}$ | 81.47 |
| DSIR | $83.07_{0.29}$ | $\mathbf{89.11}_{0.14}$ | $89.80_{0.37}$ | $\mathbf{75.09}_{2.76}$ | $90.48_{0.57}$ | $\mathbf{87.70}_{0.68}$ | $\mathbf{54.00}_{1.34}$ | $89.17_{0.13}$ | $\mathbf{82.30}$ |
| Top-$k$ DSIR | $83.39_{0.06}$ | $88.63_{0.38}$ | $\mathbf{89.94}_{0.17}$ | $72.49_{1.29}$ | $\mathbf{91.01}_{0.79}$ | $86.18_{1.12}$ | $49.90_{1.10}$ | $\mathbf{89.52}_{0.21}$ | 81.38 |
| DSIR + Neural features | $\mathbf{83.44}_{0.16}$ | $88.20_{0.35}$ | $89.81_{0.35}$ | $70.68_{2.70}$ | $90.50_{1.07}$ | $87.55_{0.99}$ | $52.58_{1.67}$ | $88.40_{0.12}$ | 81.40 |

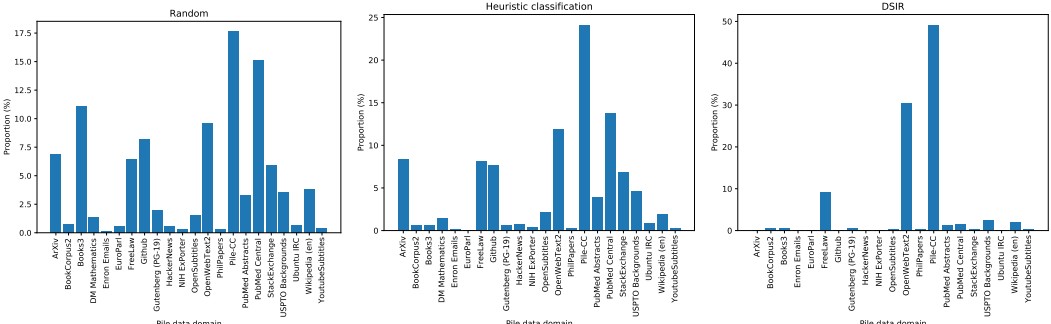

Figure 5: Distribution of Pile data sources for datasets selected by **Left:** Random selection **Middle:** Heuristic classification and **Right:** DSIR. Heuristic classification and DSIR were restricted to select only 4% of its dataset from Wikipedia, Books3, BookCorpus2, and Gutenberg.

# B    DSIR with a neural importance weight estimator

As a preliminary study, we test an instantiation of DSIR with an importance weight estimator based on neural features. For each example, we extract embeddings from a SentenceTransformer [66] (all-MiniLM-L6-v2) with dimension 384. We fit the generative models for the source and target feature space with 1000- and 50-component Gaussian mixture models respectively, with diagonal covariance structure. We use this to select pretraining data for training general-domain LMs from scratch. Table 4 shows the results using this DSIR variant in the last row. On average, DSIR with neural features improves by 1-1.5%+ over random selection and heuristic classification and is on par with top-$k$ heuristic classification and top-$k$ DSIR, but still underperforms DSIR with n-gram features. However, we believe that many aspects of this preliminary pipeline could be improved or redesigned, and that using a neural model in the importance weight estimator is a promising direction.

# C    Distribution of data sources for general-domain training

Figure 5 shows the distribution of data sources (ArXiv, GitHub, News, etc.) from The Pile that were selected by random selection, heuristic classification, and DSIR. Heuristic classification and DSIR aim to select formal text that are similar to text from Wikipedia or books. Note that we restricted heuristic classification and DSIR to select from data sources outside of Wikipedia and books sources (Books3, BookCorpus2, Gutenberg) for 96% of the dataset, while 2% is randomly selected from Wikipedia and the remaining 2% are selected from the 3 book sources. DSIR seems to focus mostly selecting formal text from web data such as Pile-CC (which can still be quite varied), while the other methods select from a variety of sources.

# D    Distribution of data sources for continued pretraining

Figure 6 shows the distribution of Pile data sources selected by DSIR for different target distributions. Each of the 4 columns represents a domain: CS papers, Biomedical text, News, and Reviews. The

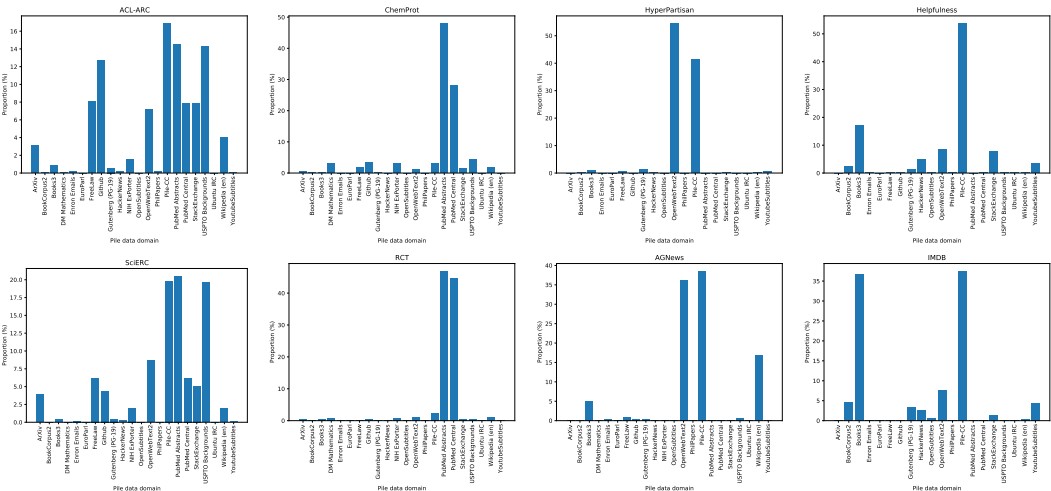

Figure 6: Distribution of Pile data sources selected by DSIR for different target distributions. The four columns from left to right represent 4 domains: CS papers, Biomedical text, News, and Reviews.

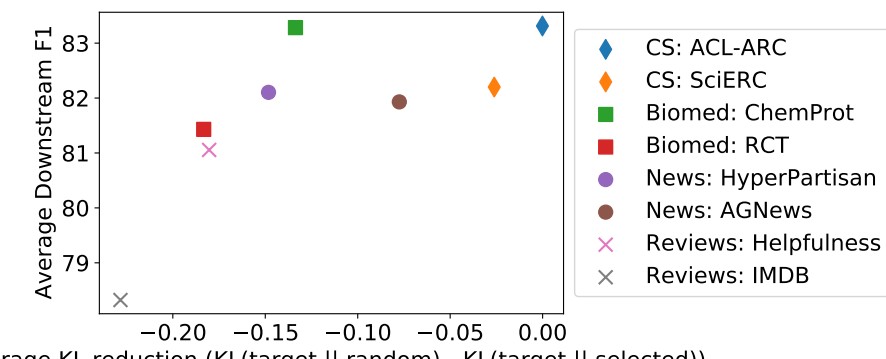

Figure 7: Plot of KL reduction against average downstream F1 score of DSIR for the 8 continued pretraining datasets, where each point represents a different pretraining dataset (selected for a particular target). Pretraining data selected for CS-related target distributions tend to transfer well to datasets in other domains, while pretraining data selected for reviews transfers poorly.

distribution of data sources for target distributions from the same domain are similar. When the target is a task from the CS domain, the distribution of data sources is the most diverse. Biomedical and news domains are particularly different; when the target is from the biomedical domain, most of the selected examples are from PubMed Abstracts and PubMed Central, and when the target is from the news domain, most of the selected examples are from web data (Pile-CC and OpenWebText2).

**Cross-domain KL reduction.** Figure 7 plots the KL reduction against average downstream F1 for all datasets selected by DSIR (one dataset for each target). KL reduction is still a strong indicator of downstream performance in this case. Data selected using CS papers tend to transfer the best to other domains, while data selected using reviews hurts performance. This also shows that transfer between domains is very asymmetric. In Figure 6, we show that the distribution of data sources selected by DSIR for CS targets is generally the most diverse, which could contributes to its strong performance on many domains. Intuitively, ACL-ARC (a dataset of NLP papers) is likely to contain a more diverse set of topics than reviews.

Table 5: Continued pretraining results on the GLUE dev set when the target distribution is formal text. DSIR improves average GLUE performance by 0.4–0.7% over all baselines. All fine-tuning results are averaged over 5 seeds. Following RoBERTa [48], for RTE, STS, and MRPC we fine-tune starting from the MNLI model instead of from scratch.

| | MNLI | QNLI | QQP | RTE | SST-2 | MRPC | CoLA | STS-B | Avg |
|---|---|---|---|---|---|---|---|---|---|
| BERT-base (no continued pretrain) | $84.29_{0.41}$ | $91.26_{0.16}$ | $90.23_{0.06}$ | $76.39_{3.80}$ | $92.34_{0.34}$ | $86.42_{2.49}$ | $56.36_{1.49}$ | $90.11_{0.23}$ | 83.43 |
| Random selection | $83.82_{0.48}$ | $89.86_{0.63}$ | $90.47_{0.39}$ | $76.03_{2.20}$ | $92.00_{0.31}$ | $87.21_{1.47}$ | $59.00_{2.57}$ | $90.32_{0.17}$ | 83.59 |
| Heuristic classification | $84.03_{0.33}$ | $90.47_{0.65}$ | $90.46_{0.36}$ | $76.75_{1.74}$ | $91.88_{0.42}$ | $86.03_{0.78}$ | $56.03_{4.22}$ | $90.30_{0.22}$ | 83.24 |
| DSIR | $84.21_{0.47}$ | $90.78_{0.42}$ | $90.45_{0.39}$ | $78.34_{1.75}$ | $92.09_{0.59}$ | $87.16_{0.77}$ | $58.41_{5.86}$ | $90.49_{0.19}$ | **83.99** |

# E   Continued pretraining results when target is formal text

We also consider using the same datasets for continued pretraining, starting from the public BERT-base checkpoint. Here, all data selection methods improve over BERT-base on the GLUE dev set. Similarly to training from scratch, we find that heuristic classification slightly decreases performance compared to random selection (by 0.2% on average). DSIR improves over random selection by 0.4% and over BERT-base by 0.6%, achieving almost 84% on the GLUE dev set.

# F   Data selection details

**Data preprocessing.**   We select data from The Pile [21], which comes in 30 random chunks. We reserve chunk 0 for validation purposes and only consider the last 29 chunks. We first divided the documents in The Pile into chunks of 128 "words", according to whitespace tokenization. These chunks define the examples that we do data selection on, totaling 1.7B examples. For heuristic classification and DSIR, we first apply a manual quality filter (Appendix J) and only consider the examples that pass the filter. Random selection selects from the unfiltered Pile.

**Heuristic classification.**   We use a bigram fasttext classification model [30], which first forms a list of unigrams and bigrams, hashes them into a predefined number of tokens (2M in this case), maps these tokens into learned feature vectors, and then learns a logistic regression model on top of averaged feature vectors across the model. We initialize the feature vectors from 300 dimensional pretrained subword fasttext vectors trained from Common Crawl. We use the fasttext hyperparameter autotuning functionality with a duration timeout of 30 minutes.

The classification model is trained on a balanced dataset of examples from The Pile validation set and examples from the target distribution (downstream unlabeled training inputs or Wikipedia/book text from The Pile validation set). We downsample the larger dataset of the two to create the balanced dataset. Each example is lowercased and stripped of newlines by first tokenizing using the NLTK word tokenizer and rejoining the words with spaces.

For noisy thresholding, we select a raw example with probability $\rho_i$ predicted by the fasttext model if $\rho_i > 1 - \beta_i$, where $\beta_i$ is sampled from a Pareto distribution with shape parameter 9. If the number of examples that do not cross the threshold is smaller than the desired number of examples $k$, then we repeat this process on the examples that were not chosen and continue to add to the dataset. After we have chosen at least $k$ examples, we take $k$ random samples without replacement from the chosen examples.

For top-$k$ heuristic classification, we simply take the examples with the top-$k$ predicted probabilities $\rho_i$.

**Importance resampling.**   Our importance resampling-based methods use a bag-of-words generative model of text. We process each example by lowercasing and splitting into words using the WordPunct tokenizer from NLTK [7]. Following [30], we incorporate unigram and bigram information by hashing the unigrams and bigrams into 10k buckets, which defines a vocabulary of 10k "words" for the generative model. Both unigrams and bigrams are hashed into the same space of words. We learn two bag-of-words models, one for the target and one for The Pile, using target data (downstream unlabeled training inputs or Wikipedia/book text from The Pile validation set) and Pile validation data. The parameters of the models are learned by simply counting the word frequencies across the dataset.

For unigram-based DSIR, we use the RoBERTa tokenizer [17], which allows us to avoid hashing. With bigrams, this is more difficult since we must consider $50000^2$ pairs of tokens in the RoBERTa vocabulary. Still, even in the unigram case we find that there are often tokens that are never seen in

Table 6: Hyperparameters for training general-domain LMs from scratch.

| | |
|---|---|
| Architecture | BERT-base |
| Max token length | 128 |
| Batch size | 4096 |
| Learning rate | 1e-3 or 8e-4 |
| Learning rate schedule | Linear |
| Weight decay | 0.01 |
| Warmup steps | 3000 |
| Total steps | 50000 |
| Optimizer | AdamW |
| Adam $\beta_1$ | 0.9 |
| Adam $\beta_2$ | 0.999 |
| Adam $\epsilon$ | 1e-8 |
| GPUs | 4 Titan RTX |

Table 7: Hyperparameters for continued pretraining of general-domain LMs.

| | |
|---|---|
| Architecture | BERT-base |
| Max token length | 512 |
| Batch size | 2048 |
| Learning rate | 1e-4 |
| Learning rate schedule | Linear |
| Weight decay | 0.01 |
| Warmup steps | 1440 |
| Total steps | 25000 |
| Optimizer | AdamW |
| Adam $\beta_1$ | 0.9 |
| Adam $\beta_2$ | 0.999 |
| Adam $\epsilon$ | 1e-8 |
| GPUs | 4 Titan RTX |

the target dataset, so we smooth the MLE parameters by mixing with the uniform distribution over tokens with a weight of 1e-5.

**Implementation of importance resampling.** We implement importance resampling with the Gumbel top-$k$ trick [81; 38; 87; 39], which produces $k$ samples without replacement according to the softmax distribution of the given scores. In the Gumbel top-$k$ procedure, we add IID standard Gumbel noise $g_i$ to each log-importance weight to produce a score $s_i = \log \frac{\hat{p}_{\text{feat}}(z_i)}{\hat{q}_{\text{feat}}(z_i)} + g_i$ for each raw example. We select the examples corresponding to the top $k$ scores. Note that producing the log-likelihood ratios and adding independent Gumbel noise to them can be trivially parallelized, and selecting top $k$ can be done in linear time with the introselect algorithm [57], implemented by `numpy.argpartition`.

**Sampling data for general-domain LMs.** To select a dataset that is suitable for both pretraining from scratch at token length 128 and continued pretraining with token length 512, we choose to first select 102.4M examples then concatenate every two examples to create 51.2M examples. This ensures that the examples are long enough for a max token length of 512 without much padding. We train the importance weight estimator or fasttext classifier from The Pile validation set, where the target is Wikipedia + BookCorpus2 + Gutenberg + Books3 and the raw data come from the rest of the data sources in The Pile. We first select 98.4M examples from non-Wikipedia and book data, then randomly select 2M from Wikipedia and 0.66M each from BookCorpus2, Gutenberg, and Books3. We mix in some examples from Wikipedia and books to balance the distribution of sources and to reduce catastrophic forgetting in continued pretraining. After this, we concatenate every two examples.

**Details for ablations.** We ablate top-$k$ heuristic classification in Section 5 in two ways. First, we consider the original heuristic classification method, which takes classifier probabilities $\rho_i = f(x_i)$ for an example and selects the example if $\rho_i > 1 - \beta_i$ where $\beta_i$ is a Pareto random variable. Second,

Table 8: Dataset-specific hyperparameters for fine-tuning LMs on GLUE, following best hyperparameters from RoBERTa [48].

|      | Epochs | Batch size | Learning rate | Continue from MNLI? |
|------|--------|------------|---------------|---------------------|
| MNLI | 10     | 32         | 1e-5          | N                   |
| RTE  | 10     | 16         | 2e-5          | Y                   |
| MRPC | 10     | 16         | 1e-5          | Y                   |
| STSB | 10     | 16         | 2e-5          | Y                   |
| COLA | 10     | 16         | 1e-5          | N                   |
| QQP  | 10     | 32         | 1e-5          | N                   |
| SST2 | 10     | 32         | 1e-5          | N                   |
| QNLI | 10     | 32         | 1e-5          | N                   |

we consider heuristic classification with importance resampling by first calibrating the classifier's probabilities with Platt scaling [63] against a validation set, then using the calibrated probabilities $\rho_i$ to compute the importance weight $\log \frac{\rho_i}{1-\rho_i}$. Similarly to DSIR, we use the Gumbel top-$k$ trick to select a subset using these importance weights.

We ablate the DSIR approach by replacing the generative importance weight estimator with a discriminative one. We use the same hashing method and define the features as 10k-dimensional counts of the n-grams. We normalize each count vector to sum to 1. On top of these features, we train a logistic regression classifier using the same dataset used to train the fasttext classifier in heuristic classification. We tune an L2 regularization weight based on best held-out accuracy (we further split the validation set in half to create another held out set) in the binary classification task. Similarly as above, we calibrate the probabilities using Platt scaling and use the classifier probabilities to compute the importance weight.

## G   Training details for training general-domain LMs

**Pretraining from scratch.**   Table 6 shows the hyperparameters for training general-domain LMs from scratch. For all models except DSIR, we use learning rate 1e-3. We use 8e-4 for DSIR since we found that 1e-3 leads to divergence in the training. We use 16 accumulation steps with 4 GPUs to achieve a large batch size of 4096, following Izsak et al. [28]. Our hyperparameters result in a compute budget of 26B tokens processed ($128 \times 4096 \times 50000$). Each training run takes about 50 hours. Our pretraining implementation is adapted from Yao et al. [89].

**Continued pretraining (Appendix E).**   Table 7 shows the hyperparameters for continued pretraining general-domain LMs. We continue pretraining from the BERT-base [17] checkpoint. During BERT training, they process 43B tokens. We process 26B tokens during training so that the total compute after continued pretraining is 69B tokens. Each continued pretraining run takes about 60 hours.

**Fine-tuning on GLUE.**   We follow the hyperparameters used by RoBERTa [48] for fine-tuning on GLUE (Tables 8 and 9). While RoBERTa searches over a space of hyperparameters, we just use the hyperparameters set for each task from the RoBERTa code base. The fine-tuning for RTE, MRPC, and STSB continues from the fine-tuned model for MNLI, following Liu et al. [48]. We use the default HuggingFace code for GLUE fine-tuning.

## H   Training details for continued pretraining of domain-specific LMs

**Pretraining.**   Table 10 shows the hyperparameters for continued pretraining domain-specific LMs. We choose the pretraining compute budget to equal the number of tokens processed in the DAPT models from Gururangan et al. [24]. For all models, we first try pretraining with learning rate 5e-4, and if training diverges, we use 1e-4.

**Fine-tuning.**   Table 11 shows the hyperparameters for fine-tuning on domain-specific datasets. We use the fine-tuning code from Gururangan et al. [24] and follow their fine-tuning protocols. For datasets from CS/Biomed/News domains, we use a max token length of 256 to match the pretraining length. For Reviews (IMDB and Helpfulness) datasets, we use a max token length of 512 since this

Table 9: Shared hyperparameters for fine-tuning LMs on GLUE, following Liu et al. [48].

| | |
|---|---|
| Architecture | BERT-base |
| Max length | 128 (from scratch) or 512 (continued pretrain) |
| Weight decay | 0.1 |
| Optimizer | AdamW |
| Adam $\beta_1$ | 0.9 |
| Adam $\beta_2$ | 0.98 |
| Adam $\epsilon$ | 1e-6 |
| Warmup ratio | 0.06 |
| LR schedule | Polynomial |
| Precision | FP16 |
| GPUs | 1 Titan RTX |

Table 10: Hyperparameters for continued pretraining on domain-specific data.

| | |
|---|---|
| Architecture | RoBERTa-base |
| Max token length | 256 |
| Total steps | 12500 |
| Batch size | 4096 |
| Weight decay | 0.01 |
| Adam $\beta_1$ | 0.9 |
| Adam $\beta_2$ | 0.999 |
| Adam $\epsilon$ | 1e-8 |
| Warmup steps | 720 |
| LR schedule | Linear |
| Learning rate | 5e-4 or 1e-4 |
| GPUs | 4 Titan RTX |

Table 11: Hyperparameters for fine-tuning on domain-specific data.

| | |
|---|---|
| Architecture | RoBERTa-base |
| Max token length | 256 or 512 |
| Epochs | 3 or 10 |
| Patience | 3 epochs |
| Batch size | 4096 |
| Weight decay | 0.1 |
| Optimizer | AdamW |
| Adam $\beta_1$ | 0.9 |
| Adam $\beta_2$ | 0.98 |
| Adam $\epsilon$ | 1e-6 |
| Warmup ratio | 0.06 |
| LR schedule | Linear |
| GPUs | 1 Titan RTX |

seems to change performance significantly. For DAPT models [24], we use a max token length of 512 for all datasets, which matches their protocol. Following Gururangan et al. [24], we choose either 3 or 10 epochs based on average validation performance over 5 seeds. Our fine-tuning implementation follows Gururangan et al. [24].

# I   Computing the KL reduction metric

To compute the KL reduction metric for a particular dataset, we took the first 100k examples from the dataset and computed the hashed n-gram counts. Normalizing these counts gives an MLE estimate of the hashed n-gram distribution for the dataset. We use the same procedure to compute the hashed n-gram distribution parameters for The Pile (from the Pile validation set).

For manual curation (DAPT), we attempted to download the datasets used in the paper (RealNews [91], S2ORC [49], and Amazon reviews [25]). However, Gururangan et al. [24] uses an internal version of S2ORC that cannot be released. We approximate S2ORC for CS papers and Biomed by using the first 100k documents in the public version of S2ORC that contain 'Computer Science' and 'Medicine' as a metadata field, respectively.

For RoBERTa, we approximate the pretraining distribution by computing the hashed n-gram distribution from Wikipedia and books data in the Pile validation set.

## J   Quality filter

For heuristic classification and IS methods, we devise a few hand-crafted ways to filter out low quality data as a preprocessing step, according to

- Word length: between 40 and 500
- Repeat ratio, defined as $\max_{\text{word}} \frac{\text{\# occurrences of word in example}}{\text{example word length}}$: between 0.02 and 0.2
- Informativeness ratio, defined as $\frac{\text{\# of non-stopwords and non-punctuation in example}}{\text{example word length}}$: between 0.3 and 0.7
- Numeric ratio, defined as $\frac{\text{\# of numbers in example}}{\text{example word length}}$: less than 0.2

The words are based on the NLTK word tokenizer [7]. These are difficult for a simple n-gram based importance weight estimator or classifier to use as features because it requires global context. We decide to keep vs. discard examples using some simple thresholds on the above values, decided using inspection on the Pile validation set. Below, we detail some statistics of the quality filtering procedure and provide some data examples.

**Statistics of quality filtering.**   With the above thresholds, we find that:

- The length filter is the most selective — after applying the length filter, only 55% of the examples are left.
- The repeat ratio filter keeps 78% of the data.
- The informativeness filter keeps 72% of the data.
- The numeric filter keeps 91% of the data.
- Overall, when applying all the filters at the same time, 52% of the examples are kept. Thus, we are mainly filtering by length, which seems like a good proxy for quality.

**Kept vs. discarded examples according to quality filter.**   First, we show the beginning characters from some randomly selected kept vs. discarded examples.

```
KEPT:
all rights, the girl should be hanged for coining and thievery, and you, sir,
millennia of ancient carvings, magical swords and glittering jewels and textiles.
Kmax, and mean asphericity ( Q) on corneal tomography were evaluated
[M]other selects a therapist who requires co-pay in\n
informations about call once you are done and you don't need info anymore

DISCARDED:
                                        (31)\\\n
SUCH DAMAGE.\n###############################################################
          +                    1
        "mpls"\n        ],\n        "setup": [\n             [\n
  1993--1997, 48 months                     NA (\\<5 yr age)
 var value = formattedTime + '\\t' + type + '\\t' + name + '\\t' + eventTxt +
                                        FILED\n
110.88 (108.42 to 113.34)   107.89 (105.28 to 110.50)   1.25 (-2.18 to 4.67)
Wye Mun no podia evitar recordar lo que su padre siempre decia: <<Nunca olvides
          2.18
bG9hdDpsZWZ0O21hcmdpbjoycHggNXB4OyB0ZXh0LWFsaWduOmNlbnRlcjsiPjxhIGhyZWY9Imh0\n
```

**Extreme length examples.**    Very short examples tend to be data snippets or otherwise nonsensical:

```
278713.303 3771574.556 by arc centered at 280828.793 3766437.062 94a to
279188.184 3771745.314 by arc centered at 280945.177 3766474.440 to 280325.491
3771995.774 by arc centered at 281478.555 3766560.741 to
```

Very long examples tend to be dense code, repetitive, or very technical:

```
$ y ' = \cap_ { h \in \mathcal { d } ( y ) \setminus g. \ { h_0 \ } } h^+ $ . the cube complex $ v = h_0^- \cap y ' $ is called a * vertebra * .
see figures \ [ fig : pentagons\ ] and \ [ vertebra\ ] . ( -4.37 , -3.17 ) rectangle ( 6.57,5.42 ) ; ( 0,0 ) - ( 0,1 ) ;
( 0,0 ) - ( 1,0 ) ; ( 1,1 ) - ( 1,0 ) ; ( 1,1 ) - ( 1,1.56 ) ; ( 0.71,1.71 ) - ( 0.85,1.71 ) ; plot\ [ domain=3.93:4.71 ,
variable=\ ] ( [ 1\ * 0.71\ * cos ( r ) +0\ * 0.71\ * sin ( r ) ] { } , [ 0\ * 0.71\ * cos ( r ) +1\ * 0.71\ * sin ( r ) ] { } ) ;
plot\ [ domain=4.71:5.5 , variable=\ ] ( [ 1\ * 0.71\ * cos ( r ) +0\ * 0.71\ * sin ( r ) ] { } , [ 0\ * 0.71\ * cos ( r ) +1\ * 0.71\
* sin ( r ) ] { } ) ; plot\ [ domain=-0.79:0 , variable=\ ] ( [ 1\ * 0.71\ * cos ( r ) +0\ * 0.71\ * sin ( r ) ] { } , [ 0\ * 0.71\
* cos ( r ) +1\ *
0.71\ * sin ( r ) ] { } ) ; plot\ [ domain=3.142:4.71 , variable=\ ] ( [ 1\ * 0.15\ * cos ( r ) +0\ * 0.15\ * sin ( r ) ] { } ,
[ 0\ * 0.15\ * cos ( r ) +1\ * 0.15\ * sin ( r ) ] { } ) ; plot\ [ domain=3.93:4.71 , variable=\ ] ( [ -1\ * 0.71\ * cos ( r )
+0\ * 0.71\ * sin ( r ) ] { } , [ 0\ * 0.71\ * cos ( r ) +1\ * 0.71\ * sin ( r ) ] { } ) ; plot\ [ domain=4.71:5.5 ,
variable=\ ] ( [ -1\ * 0.71\ * cos ( r ) +0\ * 0.71\ * sin ( r ) ] { } , [ 0\ * 0.71\ * cos ( r ) +1\ * 0.71\ * sin ( r ) ] { } ) ;
plot\ [ domain=-0.79:0 , variable=\ ] ( [ -1\ * 0.71\ * cos ( r ) +0\ * 0.71\ * sin ( r ) ] { } , [ 0\ * 0.71\ * cos ( r ) +1\ *
0.71\ * sin ( r ) ] { } ) ; plot\ [ domain=3.142:4.71 , variable=\ ] ( [ -1\ * 0.15\ * cos ( r ) +0\ * 0.15\ * sin ( r ) ] { } ,
[ 0\ * 0.15\ * cos ( r ) +1\ * 0.15\ * sin ( r ) ] { } ) ; ( 2,0 ) - ( 2,1 ) ; ( 2,0 ) - ( 1,0 ) ; ( 1,1 ) - ( 1,1.56 ) ; plot\ [
( 1.29,1.71 ) - ( 1.15,1.71 ) ; plot\ [ domain=3.93:4.71 , variable=\ ] ( [ -1\ * 0.71\ * cos ( r ) +0\ * 0.71\ * sin ( r ) ] { } ,
[ 0\ * 0.71\ * cos ( r ) +1\ * 0.71\ * sin ( r ) ] { } ) ; plot\ [ domain=4.71:5.5 , variable=\ ] ( [ -1\ * 0.71\ * cos ( r ) +0\ *
0.71\ * sin ( r ) ] { } , [ 0\ * 0.71\ * cos ( r ) +1\ * 0.71\ * sin ( r ) ] { } ) ; plot\ [ domain=-0.79:0 , variable=\ ] ( [ -1\
* 0.71\ * cos ( r ) +0\ * 0.71\ * sin ( r ) ] { } , [ 0\ * 0.71\ * cos ( r ) +1\ * 0.71\ * sin ( r ) ] { } ) ; plot\ [
domain=3.142:4.71 , variable=\ ] ( [ -1\ * 0.15\ * cos ( r ) +0\ * 0.15\ * sin ( r ) ] { } , [ 0\ * 0.15\ * cos ( r ) +1\ * 0.15\
* sin ( r ) ] { } ) ; plot\ [ domain=3.93:4.71 , variable=\ ] ( [ 1\ * 0.71\ * cos ( r ) +0\ * 0.71\ * sin ( r ) ] { } ,
[ 0\ * 0.71\ * cos ( r ) +1\ * 0.71\ * sin ( r ) ] { } ) ; plot\ [ domain=4.71:5.5 , variable=\ ] ( [ 1\ * 0.71\ * cos ( r )
+0\ * 0.71\ * sin ( r ) ] { } , [ 0\ * 0.71\ * cos ( r ) +1\ * 0.71\ * sin ( r ) ] { } ) ; plot\ [ domain=-0.79:0 , variable=\ ]
( [ 1\ * 0.71\ * cos ( r ) +0\ * 0.71\ * sin ( r ) ] { } , [ 0\ * 0.71\ * cos ( r ) +1\ * 0.71\ * sin ( r ) ] { } ) ; plot\
[ domain=3.142:4.71 , variable=\ ] ( [ 1\ * 0.15\ * cos ( r ) +0\ * 0.15\ * sin ( r ) ] { } , [ 0\ * 0.15\ * cos ( r ) +
1\ * 0.15\ * sin ( r ) ] { } ) ; ( 4,0 ) - ( 4,1 ) ; ( 4,0 ) - ( 3,0 ) ; ( 3,1 ) - ( 3,0 ) ; ( 3,1 ) - ( 3,1.56 ) ; ( 3.29,1.71 )
- ( 3.15,1.71 ) ; ( 2,0 ) - ( 3,0 ) ; ( 3,1 ) - ( 3,1.56 ) ; ( 2.71,1.71 ) - ( 2.85,1.71 ) ; plot\ [ domain=3.93:4.71 ,
variable=\ ] ( [ -1\ * 0.71\ * cos ( r ) +0\ * 0.71\ * sin ( r ) ] { } , [ 0\ * 0.71\ * cos ( r ) +1\ * 0.71\ * sin ( r ) ] { } ) ;
plot\ [ domain=4.71:5.5 , variable=\ ] ( [ -1\ * 0.71\ * cos ( r ) +0\ * 0.71\ * sin ( r ) ] { } , [ 0\ * 0.71\ * cos ( r ) +1\ *
0.71\ * sin ( r ) ] { } ) ; plot\ [ domain=-0.79:0 , variable=\ ] ( [ -1\ * 0.71\ * cos ( r ) +0\ * 0.71\ * sin ( r ) ] { } ,
[ 0\ * 0.71\ * cos ( r ) +1\ * 0.71\ * sin ( r ) ] { } ) ; plot\ [ domain=3.142:4.71 , variable=\ ] ( [ -1\ * 0.15\ *
cos ( r ) +0\ * 0.15\ * sin ( r ) ] { } , [ 0\ * 0.15\ * cos ( r ) +1\ * 0.15\ * sin ( r ) ] { } ) ; plot\ [ domain=3.93:4.71 ,
variable=\ ] ( [ 1\ * 0.71\ * cos ( r ) +0\ * 0.71\ * sin ( r ) ] { } , [ 0\ * 0.71\ * cos ( r ) +1\ * 0.71\ * sin ( r ) ] { } ) ;
plot\ [ domain=4.71:5.5 , variable=\ ] ( [ 1\ * 0.71\ * cos ( r ) +0\ * 0.71\ * sin ( r ) ] { } , [ 0\ * 0.71\ * cos ( r ) +1\ *
0.71\ * sin ( r ) ] { } ) ; plot\ [ domain=-0.79:0 , variable=\ ] ( [ 1\ * 0.71\ * cos ( r ) +0\ * 0.71\ * sin ( r ) ] { } ,
[ 0\ * 0.71\ * cos ( r ) +1\ * 0.71\ * sin ( r ) ] { } ) ; plot\ [ domain=3.142:4.71 , variable=\ ] ( [ 1\ * 0.15\ * cos ( r ) +
0\ * 0.15\ * sin ( r ) ] { } , [ 0\ * 0.15\ * cos ( r ) +1\ * 0.15\ * sin ( r ) ] { } ) ; plot\ [ domain=3.93:4.71 ,
variable=\ ] ( [ 1\ * 0.71\ * cos ( r ) +0\ * 0.71\ * sin ( r ) ] { } , [ 0\ * 0.71\ * cos ( r ) +1\ * 0.71\ * sin ( r ) ] { } ) ;
plot\ [ domain=4.71:5.5 , variable=\ ] ( [ 1\ * 0.71\ * cos ( r ) +0\ * 0.71\ * sin ( r ) ] { } , [ 0\ * 0.71\ * cos ( r ) +0\ * 0.71\ * 0.71\ *
sin ( r ) ] { } ) ; plot\ [ domain=-0.79:0 , variable=\ ] ( [ 1\ * 0.71\ * cos ( r ) +0\ * 0.71\ * sin ( r ) ] { } , [ 0\ * 0.71\ *
cos ( r ) +1\ * 0.71\ * sin ( r ) ] { } ) ; plot\ [ domain=3.142:4.71 , variable=\ ] ( [ 1\ * 0.15\ * cos ( r ) +0\ * 0.15\ *
sin ( r ) ] { } , [ 0\ * 0.15\ * cos ( r ) +1\ * 0.15\ * sin ( r ) ] { } ) ; plot\ [ domain=3.93:4.71 , variable=\ ]
( [ -1\ * 0.71\ * cos ( r ) +0\ * 0.71\ * sin ( r ) ] { } , [ 0\ * 0.71\ * cos ( r ) +1\ * 0.71\ * sin ( r ) ] { } ) ;
plot\ [ domain=4.71:5.5 , variable=\ ] ( [ -1\ * 0.71\ * cos ( r ) +0\ * 0.71\ * sin ( r ) ] { } , [ 0\ * 0.71\ * cos ( r ) +
1\ * 0.71\ * sin ( r ) ] { } ) ; plot\ [ domain=-0.79:0 , variable=\ ] ( [ -1\ * 0.71\ * cos ( r ) +0\ * 0.71\ * sin ( r ) ] { } ,
[ 0\ * 0.71\ * cos ( r ) +1\ * 0.71\ * sin ( r ) ] { } ) ; plot\ [ domain=3.142:4.71 , variable=\ ] ( [ -1\ * 0.15\ * cos ( r )
+0\ * 0.15\ * sin ( r ) ] { } , [ 0\ * 0.15\ * cos ( r ) +1\ * 0.15\ * sin ( r ) ] { } ) ; ( 8,0 ) - ( 8,1 ) ; ( 8,0 ) - ( 7,0 ) ;
( 7,1 ) - ( 7,0 ) ; ( 7,1 ) - ( 7,1.56 ) ; ( 7.29,1.71 ) - ( 7.15,1.71 ) ; ( 6,0 ) - ( 6,1 ) ; ( 6,0 ) - ( 7,0 ) ; ( 7,1 ) -
( 7,1.56 ) ; ( 6.71,1.71 ) - ( 6.85,1.71 ) ; ( 4,0 ) - ( 5,0 ) ; ( 5,1 ) - ( 5,0 ) ; ( 5,1 ) - ( 5,1.56 ) ; ( 4.71,1.71 ) -
( 4.85,1.71 ) ; ( 6,0 ) - ( 5,0 ) ; ( 5,1 ) - ( 5,1.56 ) ; ( 5.29,1.71 ) - ( 5.15,1.71 ) ; plot\ [ domain=3.93:4.71
, variable=\ ] ( [ -1\ * 0.71\ * cos ( r ) +0\ * 0.71\ * sin ( r ) ] { } , [ 0\ * 0.71\ * cos ( r ) +1\ * 0.71\ * sin ( r ) ] { } ) ;
```

**Extreme repeat ratio example.**    Examples with a high repeat ratio are mostly examples without much content except for one repeated token, sometimes within code:

```
$ d_h ( x , y+\delta ) $ -- -- -- -- -- -- -- -- -- -- -- -- -- --
-- -- -- -- -- -- -- -- -- -- -- -- -- -- -- -- -- -- -- -- -- --
-- -- -- -- -- -- -- -- -- -- -- -- -- -- -- -- -- -- -- -- -- --
-- -- -- -- -- -- -- -- -- -- -- -- -- -- -- -- -- -- -- -- -- --
-- -- -- -- -- -- -- -- -- -- -- -- -- -- -- -- -- -- -- -- -- --
-- -- -- -- -- -- $ \left ( -\infty , \frac { 3\delta } { 4 }
+\eps\delta\right ) $ $ ( x_0 , y ) $ -

-- -- -- -- -- -- -- -- -- -- -- -- -- -- -- -- -- -- -- -- -- --
-- -- -- -- -- -- -- -- -- -- -- -- -- -- -- -- -- -- -- -- -- --
-- -- -- -- -- -- -- -- -- -- -- -- -- -- -- -- -- -- -- -- -- --
-- -- - -- -- -- -- -- -- -- - -- -- -- -- -- -- -- - -- -- -- --
-- -- -- - -- -- -- -- -- -- - -- --
* * i look forward to one day becoming a mother * * * * 0 * * .
* * 871 * * * * 0 * * . * * 870 * * -0.130 -0.312 0.066 -0.286
```

**Extreme informativeness ratio examples.**    Very low informative ratio examples also tend to be (sometimes extremely) short:

| |

Extremely high informativeness examples are often foreign language, since they don't contain English stop words:

```
maailmankaikkeuden sinnikkäimmällä valittajahahmolla umayya abu-hannalla on taas
sanottavaa . niin , tiedätte kyllä mistä : suomalaisten ra-sis-mis-ta .
abu-hannaa haastatellaan päivän helsingin sanomissa ehkä ziljoonannen kerran
tästä yhdestä ja samasta aiheesta . en ymmärrä miksi . abu-hanna on ollut
tällä viikolla suomessa noutamassa global family award -palkintoa ''
avarakatseisuudesta ja monikulttuurisuuden merkittävästä edistämisestä
suomalaisessa yhteiskunnassa '' . avarakatseisuudesta ? ? ? ? ?
en tiedä ketään , jonka katsantokanta on ( julkisen kuvan perusteella ) niin kapea
kuin abu-hannan . hänen toistuvat '' suuri osa suomalaisista on rasisteja ''
-puheenvuoronsa eivät myöskään synnytä minkäänlaista positiivista dialogia tähän
yhteiskuntaan . niistä tykkäävät ja somessa peukuttavat ainoastaan ne ihmiset ,
jotka ovat itsekin sitä mieltä , että suomi on raakalaismaisten rasistien maa .
muissa suomalaisissa ne herättävät kohtuuttomuudessaan ja jankkauksessaan vain
ärtymystä . kuten kaikki tiedämme , abu-hanna asuu nykyään hollannissa . vielä
vuosi sitten kyseinen maa näyttäytyi hänen haastatteluissaan paratiisina . mutta
nyt - ja tämä ei varmasti yllätä ketään -
```

Examples with informative ratio close to 0.5 are more standard English:

```
the feature ( 2 ) mentioned above . the assumption of $ p $ = 1 might
be unrealistic in  usual ferromagnetic metals . however , if the exchange
interaction between eu atoms is  accomplished via $ \pi $ -bands of
c $ _ { 60 } $ as discussed earlier , we can expect a  large spin
polarization of $ \pi $ -electrons . we can also consider the effect of magnetic
polaron . in magnetic semiconductors such as eu chalcogenides ,
a carrier makes surrounding magnetic moments be polarized via
exchange interaction and forms a magnetic polaron [ @ kasuya1 ] .
at zero field , magnetic polarons have to move with flipping some
magnetic moments which are more or less randomly oriented , and their
conduction is suppressed . application of magnetic field aligns spin
directions and carriers become mobile . as a result , negative
magnetoresistance occurs . the negative magnetoresistance above
$ t_c $ can be attributed to this
```

