# OpenReview forum: "Data Selection for Language Models via Importance Resampling"
_NeurIPS.cc/2023/Conference — NeurIPS 2023 poster_

### Official Review · Reviewer_LQ7V · 2023-06-28

**Soundness:** 2 fair
**Presentation:** 3 good
**Contribution:** 3 good
**Rating:** 6
**Confidence:** 4

**Summary:**

This paper aims to improve the samples to pre-train language models.
It propose a novel method called "Data Selection with Importance Resampling (DSIR)" to select better pre-training samples and found superior fine-tuning performance compared to various other approaches, including random-sampling.
The authors show the effectiveness of this approach by reporting 2-2.5% better average performance on the GLUE dataset.

**Strengths:**

This paper addresses a crucial issue "how can we make pre-training more efficient?"
It is well written, easy to follow, covers relevant related work, and considers various other methods.
Their approach effectively improves performance when fine-tuning models on downstream tasks.

**Weaknesses:**

- While show performance improvement, this paper considers GLUE a rather old, general language benchmark. Therefore, this paper evaluates only a part of the possible tasks. Especially in these times when we see the big success of larger language models on such general tasks using in-context learning, it would be interesting to see how models perform on other more challenging tasks - like reasoning intense ones.
- This paper only reports the GLUE performance on the dev set. While we see an improvement, it remains to be seen how this transfer to the test set. This is especially relevant since we do not know whether the authors use the dev set to find the best-performing epoch. From the paper, we know that RoBERTa default hyperparameters were used - which includes early stopping on the dev set - but in this paper nothing is stated regarding this issue.
- The authors report that domain-specific pre-training heavily harms downstream tasks' performance in the worst case. This also reflects a crucial limitation of this approach; when pre-training a language model, we aim for an optimal model in general without having specific downstream tasks/domains in mind. But with this work, we introduce new dependence to the target domain, which only sometimes matches the full richness of existing and upcoming language properties/domains.

**Questions:**

- How do you see your approach connected to curriculum learning, where we gradually increase the difficulties during pre-training?
- Do you think more recent learning paradigms like in-context learning or prompting can gain in the same way as fine-tuning does?

**Limitations:**

Limitations are addressed in the paper.

---

> ### Author Rebuttal · Authors · 2023-08-09
>
> We thank LQ7v for the feedback. Overall, LQ7v felt that the paper was “well written” and “addresses a crucial issue”. We answer specific questions below:
>
> > “we do not know whether the authors use the (GLUE) dev set to find the best-performing epoch. From the paper, we know that RoBERTa default hyperparameters were used - which includes early stopping on the dev set - but in this paper nothing is stated regarding this issue.”
>
> **To clarify, we do not use the dev set for tuning hyperparameters and instead use a fixed set of hyperparameters** for each GLUE dataset. By the RoBERTa default hyperparameters, we mean that we copied and fixed the default hyperparameters for learning rate, batch size, and number of epochs provided in the RoBERTa official codebase, without using any of their additional tuning strategies. We apologize for any confusion and will clarify in the final revision.
>
> >  “this paper considers GLUE a rather old, general language benchmark…it would be interesting to see how models perform on other more challenging tasks - like reasoning intense ones.”
>
> Due to compute limitations, we weren’t able to pretrain a model that is large enough for reasoning capabilities from scratch for the general-domain experiments. However, we agree that challenging reasoning tasks are a great direction for future evaluation of data curation methods on large LMs and we would like to do this in the future. We will add this in the discussion for the final revision.
>
> > “when pre-training a language model, we aim for an optimal model in general without having specific downstream tasks/domains in mind. But with this work, we introduce new dependence to the target domain…”
>
> - In general, the target distribution is a hyperparameter that can be tuned to be as general as we desire. For example **in the general-domain experiments we show that even a heuristic/proxy target distribution (Wikipedia and books) can improve performance on general downstream benchmarks.** For general-domain models, the task then becomes to design a good target distribution for high-quality data, and then apply DSIR.
> - For **domain-specific models, we often have a target distribution of interest** (e.g., code, law, medical) and DSIR can be applied directly to gather more relevant data for this target distribution.
>
> > Do you think more recent learning paradigms like in-context learning or prompting can gain in the same way as fine-tuning does?
>
> - **Yes.** For example, the GLaM paper (https://arxiv.org/abs/2112.06905) shows that **even heuristic classification (filtering the pretraining data with a Wikipedia classifier) can significantly improve few-shot in-context learning** performance. We believe **DSIR is a more principled approach and would bring further gains** at these large scales.
> - There are also some works that show that in-context learning benefits from selecting the few-shot examples themselves intelligently (https://aclanthology.org/2022.emnlp-main.622/, https://arxiv.org/abs/2302.13539, https://arxiv.org/pdf/2307.07164.pdf, https://arxiv.org/abs/2101.06804, https://arxiv.org/abs/2301.11916).
> - Another related area is selecting good examples for RLHF / instruction tuning of large models. The target distribution here could roughly be described as “helpful answers to a diverse set of prompts”. DSIR could be used to scale up the amount of good instruction tuning data for RLHF.
>
> > How do you see your approach connected to curriculum learning, where we gradually increase the difficulties during pre-training?
>
> Curriculum learning could be viewed within the DSIR framework as selecting data against a set of target distributions that changes over the course of training, starting with preferring easier examples and progressing to harder ones.

---

> > ### Author Response · Authors · 2023-08-14
> > **Further discussion**
> >
> > Dear reviewer, may we ask if you could respond to our comments? In our response, we clarify that we do not use the GLUE validation set to tune hyperparameters, address the generality of having a target distribution, and address other detailed concerns. Please let us know if you have other questions or concerns. Thank you!

---

> > > ### Comment · Reviewer_LQ7V · 2023-08-16
> > >
> > > Thanks a lot of taking the time to answer my questions and addressing raised concerns. The response provide valuable feedback and insights, therefore I'll increase my scores.
> > >
> > > Regarding the raised concern of the hyperparameters.
> > > Thanks a lot of clarifying. Nevertheless, according to the RoBERTa paper, they tuned these parameters for fine-tuning on the dev set. Therefore, having the test set scores on GLUE would be more convincing.

---

### Official Review · Reviewer_znTv · 2023-07-03

**Soundness:** 3 good
**Presentation:** 4 excellent
**Contribution:** 4 excellent
**Rating:** 8
**Confidence:** 5

**Summary:**

The work presents a novel framework for effectively selecting a representative document subset during the pre-training of Large Language Models. Pre-training such models incurs significant costs, prompting efforts to minimize the subset size and associated expenses. The authors commence by highlighting the importance of the subject matter and its practical applications. Subsequently, they identify the limitations and characteristics of existing approaches. Building upon the recognized limitations and compelling evidence of Importance Resampling's applicability in the context of Large Models, the authors propose an innovative framework based on KL reduction.

In general, the paper has a good idea and a good novelty factor. The authors claim they improved the state-of-the-art on the text classification task, and, indeed, there is strong evidence by the results presented in the paper. In sum, my unique improvement suggestion is to include statistical treatment of the presented results.

In summary, this research exhibits a commendable goal and innovative ideas, demonstrating substantial potential. With minor changes addressed, I recommend accepting this paper. I would like to extend my congratulations to the authors for their extensive experimental work and the promising results they have achieved.


**Strengths:**

S1: The paper is well written. The authors clearly define the evaluated objectives, motivation, and contributions.

S2: The implementation details and method-specific hyperparameters were defined. Thus, the paper is (possibly) reproducible. Besides, the authors shared their code as supplementary material with the submission in the OpenReview.

S3: The considered datasets are well-known and widely used in the literature. ACL, Sci-ERC, ChemProt, RCT, AGNews, HyperPartisan, Helpfulness, and IMDB.

S4: The proposed method was fairly compared to strong baselines. Besides, in terms of text classification methods, the proposed method was applied to RoBERTa, a strong SOTA method in the LLM field. Moreover, the baselines were very well explained in a simple and straightforward way.

S5: The authors adopted proper metrics to handle and properly measure the effectiveness on both balanced and unbalanced datasets domains (Accuracy and Macro-F1, respectively).

S6: Repetition: It is worth noting that the authors adopted a 5-Fold validation procedure, which demonstrates their commitment to rigorously evaluating the proposed framework.


**Weaknesses:**

W1: There is no statistical treatment of the results (e.g., statistical significance tests), which does not allow to rule out the null hypothesis of equality of results. There are strong evidences of the superiority. However, without tests, any claim of superiority can be considered unsubstantiated. To strengthen the research claims and ensure robust conclusions, it is desirable to include appropriate statistical analyses to validate the significance of the reported results.

**Questions:**

Q1: Overall, the paper demonstrates a strong conceptual framework with a commendable novelty factor. However, there is room for improvement, particularly in addressing the previously mentioned weakness.

Regarding this weakness, it is worth noting that the authors have already conducted a 5-Fold random validation procedure. As a result, incorporating a statistical method to strengthen the research claims would require minimal additional effort. In light of this, I highly recommend applying a t-test with Bonferroni correction to account for multiple tests. [1,2]

[1] Dacrema, M. F., Cremonesi, P., & Jannach, D. (2019). Are we really making much progress? A worrying analysis of recent neural recommendation approaches. In Proceedings of the 13th ACM conference on recommender systems (pp. 101–109).

[2] Cunha, Washington, et al. "On the cost-effectiveness of neural and non-neural approaches and representations for text classification: A comprehensive comparative study." Information Processing & Management 58.3 (2021): 102481

---

> ### Author Rebuttal · Authors · 2023-08-09
>
> We thank the reviewer for the feedback. Overall, znTv notes that we provide **“a strong conceptual framework with a commendable novelty factor”, along with “extensive experimental work and promising results”**. We respond to specific questions below:
>
> > “There are strong evidences of the superiority… To strengthen the research claims and ensure robust conclusions, it is desirable to include appropriate statistical analyses to validate the significance of the reported results… I highly recommend applying a t-test with Bonferroni correction to account for multiple tests.”
>
> The statistical tests below show that **DSIR outperforms all other methods with significance (α=0.05) in general-domain pretraining and is comparable with manual expert curation / top-k heuristic classification in domain-specific continued pretraining while outperforming all other baselines with significance. This is in line with what is claimed in the paper.** We conducted Student-t tests (unequal variances) on the average performance of the methods across downstream datasets, with a Bonferroni correction for multiple comparisons between DSIR and other methods. We thank the reviewer for the suggestion and will include it in the final revision.
>
> For general-domain pretraining, with p-value threshold 0.05 / 3 = 0.0125:
>
> |                             | t-score     | p-value         |
> |-----------------------------|-------------|-----------------|
> | DSIR vs random              | 9.271188391 | 0               |
> | DSIR vs heuristic cls       |  10.8322228 | 0               |
> | DSIR vs top-k heuristic cls | 3.716599729 | 0.0001 |
>
> For domain-specific continued pretraining, with p-value threshold 0.05 / 5 = 0.01:
>
> |                             | t-score      | p-value            |
> |-----------------------------|--------------|--------------------|
> | DSIR vs RoBERTa             |  5.064870983 | 0.0000002|
> | DSIR vs manual curation     |  0.816252706 |       0.207 |
> | DSIR vs random              |  3.199222061 |    0.0006 |
> | DSIR vs Heuristic cls       |  2.987762708 |     0.0014 |
> | DSIR vs Top-k heuristic cls | 0.2570449689 |       0.398 |

---

> > ### Comment · Reviewer_znTv · 2023-08-14
> >
> > Thanks for the replies. Considering that the authors have addressed my particular suggestion and have integrated the suggested statistical analysis, it seems they've proactively enhanced the robustness of their research and substantiated their conclusions. In light of this, I'll maintain my Strong Accept rating.

---

### Official Review · Reviewer_rytZ · 2023-07-06

**Soundness:** 4 excellent
**Presentation:** 4 excellent
**Contribution:** 4 excellent
**Rating:** 8
**Confidence:** 3

**Summary:**

The authors present a novel framework for selecting examples from a large and diverse dataset which are most relevant to a specific target domain. They apply this towards data selection for 'continued pretraining' in which a pretrained LM is trained further on a domain-specific dataset, in order to improve its ultimate downstream performance on domain-relevant tasks. They also define a novel data-evaluation metric which is cheap to calculate and correlates well with downstream task performance. They evaluate the proposed method on continued training for RoBERTa against competing methods for 8 tasks, and demonstrate superior average performance. They present additional experiments ablating the target domain dataset, and demonstrate the utility of the method towards data selection for a 'general domain' LM (via heuristic of formality).

**Strengths:**

This paper was a pleasure to read. The overall problem addressed in the paper is important and especially relevant given the preponderance of general LLMs in contemporary research. The relative tractability of this the proposed method at scale makes it more pragmatic than some competing approaches. The paper is clearly written and well structured, each section flows nicely from the last. The results outperform extant methods, and are thoroughly and engagingly analyzed. The proposed method provides a tractable and approachable means of solving a quite general problem, as well as presents many compelling directions for future work in a similar direction.

**Weaknesses:**

DSIR is general to the kind of feature extractor in use, yet in this work only hashed n-gram and unigram feature are explored. It would be instructive as to demonstrate DSIR with different feature spaces. Naively I would assume this is a more interesting ablation than discriminative vs generative approach to learning the importance weight estimator, as this also might demonstrate a tradeoff of performance and efficiency. However, n-grams are simple and a very reasonable place to start. This does not significantly detract from the overall work, and this omission is rightly mentioned in the limitations as being left for future work.

Nits:
Figure 3 and Table 3 cover the same data, ideally would be presented on the same page.
Table 2 and its analysis (lines 199-212) are also on different pages.
Figure 3 references the overall avg performance of the values reported in Table 3, please add that as a final column to the Table itself. Additionally, please add a separate row showing the performance of the baseline model with no continued training.
Table 3 might be made more readable (depending on execution) by color-coding the relative performance each dataset. As presented, it is very difficult to parse that ChemProt is stronger on avg than Sci-ERC, that IMDB is weak on average, or that IMDB on HyperPartisan is such a negative outlier. Additionally, it might be useful to standardize the column widths and center the text so the main diagonal is more obviously a diagonal.

**Questions:**

In Table 3, the strong performance of the CS-dataset-target models on non-target domains, especially compared to the relatively poor performance of the Reviews datasets, shows asymmetry in the transfer between domains (line 225). This is an interesting finding and could bear slightly more explanation.

In the discussion, it is mentioned (line 319) that amplifying biases in the target data is a potential harm. On the other hand, the possibility of mitigating bias might confer a tremendous benefit; what do the authors think of the potential to use DSIR as a means of removing bias from a giant dataset?

**Limitations:**

The limitations of the work and potential negative societal impact are sufficiently addressed.

---

> ### Author Rebuttal · Authors · 2023-08-09
>
> We thank the reviewer for the feedback. Reviewer rytZ felt that the problem is **“important and especially relevant”** and the paper provides a **“tractable and approachable means of solving a quite general problem, as well as presents many compelling directions for future work”**. We answer specific questions below:
>
> > DSIR is general to the kind of feature extractor in use, yet in this work only hashed n-gram and unigram feature are explored. It would be instructive as to demonstrate DSIR with different feature spaces.
>
> **During the rebuttal period, we implemented a first-pass version of DSIR with embeddings from a pretrained language model.** We extracted features using a Sentence Transformer (miniLM-v6-2), learned raw and target distributions over the features parameterized as 1000 and 50-component Gaussian mixture models respectively, and used these within the DSIR framework to select data for general-domain pretraining. The results are shown below in a table. **On average, DSIR with neural features improves by 1-1.5%+ over random selection and heuristic classification** and is on par with top-k heuristic classification and top-k DSIR, but still underperforms DSIR with n-gram features. However, we believe that this is still a promising direction since some steps could be improved, such as the hyperparameters of the Gaussian mixture model. We thank the reviewer for the suggestion.
>
>
> | GLUE dev                       |  MNLI |  QNLI |   QQP |   RTE | SST-2 |  MRPC |  CoLA | STS-B |      Avg |
> |--------------------------------|------:|------:|------:|------:|------:|------:|------:|------:|---------:|
> | Random selection               | 82.63 |  86.9 | 89.57 | 67.37 | 90.05 |  87.4 | 49.41 | 88.63 |   80.25 |
> | Heuristic classification       | 82.69 | 85.95 | 89.77 | 68.59 | 88.94 | 86.03 | 48.17 | 88.62 |   79.85 |
> | Top-k heuristic classification | 83.34 | 88.62 | 89.89 | 70.04 | 91.15 | 86.37 | 53.02 |  89.3 | 81.47 |
> | Top-k DSIR                     | 83.39 | 88.63 | 89.94 | 72.49 | 91.01 | 86.18 |  49.9 | 89.52 |  81.38 |
> | DSIR + n-gram features                          | 83.07 | 89.11 |  89.8 | 75.09 | 90.48 |  87.7 |    54 | 89.17 |  82.30 |
> | DSIR + neural features         | 83.44 |  88.2 | 89.81 | 70.68 |  90.5 | 87.55 | 52.58 |  88.4 |   81.40 |
>
>
> > In Table 3, the strong performance of the CS-dataset-target models on non-target domains, especially compared to the relatively poor performance of the Reviews datasets, shows asymmetry in the transfer between domains (line 225). This is an interesting finding and could bear slightly more explanation.
>
> In Appendix Figure 6, we show that the distribution of data sources selected by DSIR for CS-dataset targets is generally much more diverse, and we believe this could be a reason for its strong performance on many domains. This might make sense since ACL-ARC (a dataset of NLP papers) is likely to be on a more diverse set of topics than reviews.
>
> > the possibility of mitigating bias might confer a tremendous benefit; what do the authors think of the potential to use DSIR as a means of removing bias from a giant dataset?
>
> **DSIR has potential for removing dataset bias by collecting a target dataset with less bias** and using it to define the target distribution. For example, if the target dataset has roughly equal representation across groups, DSIR will upweight minority groups and downweight majority groups accordingly. However, since the user interface is simply to collect a target dataset (and no groups have to be explicitly defined), more intricate patterns about what makes the target dataset less biased can also be taken into account. We thank the reviewer for the great idea!
>
> > Nits / presentation of tables and figures
>
> We will update these in the final revision - we thank the reviewer for the suggestions.

---

> > ### Comment · Reviewer_rytZ · 2023-08-14
> >
> > Thanks for the replies. I maintain my Strong Accept rating.

---

### Official Review · Reviewer_NB9k · 2023-07-07

**Soundness:** 2 fair
**Presentation:** 3 good
**Contribution:** 2 fair
**Rating:** 4
**Confidence:** 4

**Summary:**

This paper proposes a simple method for selecting pretraining data for language modeling based on the downstream fine-tuning tasks. It uses ngrams as features for a corpus, and weighs the pretraining data based on importance sampling, which estimates how similar the pretraining data is to the task specific fine-tuning data.

**Strengths:**

1. The method is simple and easy to understand. The presentation of the paper is generally easy to follow
2. The paper tested the method on both domain adaptive continued pretraining and training from scratch


**Weaknesses:**

1. The paper uses a simple n-gram feature for importance weighting, but it’s not clear whether other dense features would bring better performance. Even if using dense features is more expensive or unrealistic, it would be helpful to have a comparison on the small scale, or analysis of the differences in computation required for using different features.
2. The paper did not compare to other automatic data selection methods such as
3. The method does seem like a simple/scalable approach, but there is not a good analysis on the resources(memory/training time) needed.

**Questions:**

1. Have you considered comparing to learned data selection method such as DoReMi(https://arxiv.org/pdf/2305.10429.pdf)?

---

> ### Author Rebuttal · Authors · 2023-08-09
>
> We thank the reviewer for their feedback. Reviewer NB9k notes that the method “seems like a simple/scalable approach” and the paper included comprehensive experimental settings, as it “tested the method on both domain adaptive continued pretraining and training from scratch”. We answer specific questions below:
>
> > “Even if using dense features is more expensive or unrealistic, it would be helpful to have a comparison on the small scale”
>
> **During the rebuttal period, we implemented a first-pass version of DSIR with embeddings from a pretrained language model.** We extracted features using a Sentence Transformer (miniLM-v6-2), learned raw and target distributions over the features parameterized as 1000 and 50-component Gaussian mixture models respectively, and used these within the DSIR framework to select data for general-domain pretraining. The results are shown below in a table. **On average, DSIR with neural features improves by 1-1.5%+ over random selection and heuristic classification** and is on par with top-k heuristic classification and top-k DSIR, but still underperforms DSIR with n-gram features. However, we believe that this is still a promising direction since some steps could be improved, such as the hyperparameters of the Gaussian mixture model. We thank the reviewer for the suggestion.
>
>
> | GLUE dev                       |  MNLI |  QNLI |   QQP |   RTE | SST-2 |  MRPC |  CoLA | STS-B |      Avg |
> |--------------------------------|------:|------:|------:|------:|------:|------:|------:|------:|---------:|
> | Random selection               | 82.63 |  86.9 | 89.57 | 67.37 | 90.05 |  87.4 | 49.41 | 88.63 |   80.25 |
> | Heuristic classification       | 82.69 | 85.95 | 89.77 | 68.59 | 88.94 | 86.03 | 48.17 | 88.62 |   79.85 |
> | Top-k heuristic classification | 83.34 | 88.62 | 89.89 | 70.04 | 91.15 | 86.37 | 53.02 |  89.3 | 81.47 |
> | Top-k DSIR                     | 83.39 | 88.63 | 89.94 | 72.49 | 91.01 | 86.18 |  49.9 | 89.52 |  81.38 |
> | DSIR + n-gram features                          | 83.07 | 89.11 |  89.8 | 75.09 | 90.48 |  87.7 |    54 | 89.17 |  82.30 |
> | DSIR + neural features         | 83.44 |  88.2 | 89.81 | 70.68 |  90.5 | 87.55 | 52.58 |  88.4 |   81.40 |
>
>
> > “there is not a good analysis on the resources(memory/training time) needed”, and it would be helpful to have an “analysis of the differences in computation required for using different features.”
>
> - **DSIR is compute and memory efficient due to the simple n-gram featurization.** The memory usage is constant (keep track of 10k n-gram frequency counts), and it takes about 38 minutes to compute n-gram counts on the Pile (2B examples) using 30 workers.
> - **Comparison between n-gram and neural features: using neural embeddings requires at least 138M times more FLOPs+integer ops than the n-gram featurization, if a small 23M parameter neural embedding model.** This is mainly due to the FLOPs needed to run the forward pass of the neural embedding model vs. counting n-grams. **Due to the compute costs, it takes about 16 hours to extract neural embeddings from the Pile with 30 workers with 1 GPU each (23M model), vs 38 minutes with n-gram features**
> - We thank the reviewer for the comment and will add more discussion of the compute resources needed in the final revision.
>
> > “Have you considered comparing to learned data selection method such as DoReMi(https://arxiv.org/pdf/2305.10429.pdf)?”
>
> - DSIR and DoReMi tackle different data curation problems. **DoReMi does not select data for a target distribution and can only reweight the data on the coarser level of domains** (instead of example level selection). Instead, DoReMi tries to find a reweighting that is robust to many target distributions via minimax optimization. Although DoReMi could be applicable to the general-domain experiments, scaling up DoReMi to example level selection (where domain = 1 example) would require some fundamental changes to DoReMi, since it currently depends on seeing all/most domains in the minibatch to keep the domain weights updated.
> - We also note that **DoReMi was also released on arXiv after the NeurIPS deadline**.

---

> > ### Author Response · Authors · 2023-08-14
> > **Further discussion**
> >
> > Dear reviewer, may we ask if you could respond to our comments? In our response, we explore the use of neural features and clarify the memory/time resources for the data selection method. Please let us know if you have other questions or concerns. Thank you!

---

> > > ### Author Response · Authors · 2023-08-20
> > > **Request for further discussion**
> > >
> > > Dear reviewer, please let us know if we've addressed your concerns regarding use of dense features and the memory/time resources. We are happy to answer any other questions you may have. Thank you!

---

> > > > ### Comment · Reviewer_NB9k · 2023-08-22
> > > > **Rebuttal response**
> > > >
> > > > Thanks for adding the additional experiments! I think these numbers would make the paper more complete and useful for readers. I do think the paper would be way more stronger if the comparison with other dense embedding based method is emphasized, so that the main motivation would be to reduce cost for data selection. It is hard to improve the current score because these results would lead to pretty big changes to the paper. Therefore, I maintain my current score but would really like to see the paper revised to reflect this new finding.

---

### Official Review · Reviewer_LSWj · 2023-07-07

**Soundness:** 3 good
**Presentation:** 2 fair
**Contribution:** 2 fair
**Rating:** 4
**Confidence:** 4

**Summary:**

The authors begin by highlighting an important issue related to data quality in (encoder only) LMs, motivating the need for an improved way to filter large datasets (e.g. The Pile) for samples which are in distribution to a held out target sample. The authors propose a metric, KL-reduction, which quantifies how well matching a target distribution signals downstream performance after fine-tuning on the data. To sample in-distribution points, they compute a importance weight for each sample, by learning a generative distribution for the raw data and target data. Finally they re-sample data using the weights without replacement. Beyond relying on the KL-reduction, the authors finetune RoBERTa on their filtered data (compared to other baseline filtering/sampling methods) and measure performance improvements on the downstream task, showing several percentage point improvements.

**Strengths:**

* Paper is well written and easy to understand; motivation and set up are clear.
* The described KL-reduction is a simple metric which appears to strongly correlate with downstream performance after fine-tuning, saving time and compute costs.
* Paper shows strong performance improvements on existing datasets/baselines using the proposed strategy


**Weaknesses:**

* The authors only run experiments using Encoder only models, this is in line with a 2021 paper which their benchmarks (data + a model) are based on. However given the recent attention to decoder only models, and the shown impact data quality has on pretraining (see llama / red pajama), an obvious question is if decoder only models also benefit from this low cost strategy. Including decoder/generative experiments may increase the impact of the method.
* The task shares similarities with active learning, that should include be included somewhere in the related work section.
* The authors only consider n-gram based features, it’s mentioned in the limitations but it likely wouldn’t take much to run the experiments with other features. On Line 173, the authors mention the potential for BERT embeddings, but there are no related experiments. Clearly, the use of just n-grams is effective, but why are n-grams the best way to determine if something is ‘in distribution’ with a target text?


**Questions:**

* TF-IDF (or bm25), i.e. a simple retrieval system, could be used to retrieve documents relevant to the target set sufficient. Line 263 references experiments but the results should be included. Related to this, if a bm25 approach will yield many duplicate documents, why can’t they just be deduplicated? And why does the proposed approach not yield duplicates?

* How are the distributions (q(x) and p(x)) on lines 91/92 computed? Elsewhere it is described as a generative distribution, but is it simply parameterized by counting n-grams in the respective datasets?


**Limitations:**

The discussion of limitations is adequate.

---

> ### Author Rebuttal · Authors · 2023-08-09
>
> We thank the reviewer for the feedback. LSWj notes that the paper is **“highlighting an important issue” and “shows strong performance improvements on existing datasets/baselines”**. We answer specific questions below:
>
> > “The authors only consider n-gram based features, it’s mentioned in the limitations but it likely wouldn’t take much to run the experiments with other features”
>
> **During the rebuttal period, we implemented a first-pass version of DSIR with embeddings from a pretrained language model.** We extracted features using a Sentence Transformer (miniLM-v6-2), learned raw and target distributions over the features parameterized as 1000 and 50-component Gaussian mixture models respectively, and used these within the DSIR framework to select data for general-domain pretraining. The results are shown below in a table. **On average, DSIR with neural features improves by 1-1.5%+ over random selection and heuristic classification** and is on par with top-k heuristic classification and top-k DSIR, but still underperforms DSIR with n-gram features. However, we believe that this is still a promising direction since some steps could be improved, such as the hyperparameters of the Gaussian mixture model. We thank the reviewer for the suggestion.
>
> | GLUE dev                       |  MNLI |  QNLI |   QQP |   RTE | SST-2 |  MRPC |  CoLA | STS-B |      Avg |
> |--------------------------------|------:|------:|------:|------:|------:|------:|------:|------:|---------:|
> | Random selection               | 82.63 |  86.9 | 89.57 | 67.37 | 90.05 |  87.4 | 49.41 | 88.63 |   80.25 |
> | Heuristic classification       | 82.69 | 85.95 | 89.77 | 68.59 | 88.94 | 86.03 | 48.17 | 88.62 |   79.85 |
> | Top-k heuristic classification | 83.34 | 88.62 | 89.89 | 70.04 | 91.15 | 86.37 | 53.02 |  89.3 | 81.47 |
> | Top-k DSIR                     | 83.39 | 88.63 | 89.94 | 72.49 | 91.01 | 86.18 |  49.9 | 89.52 |  81.38 |
> | DSIR + n-gram features                          | 83.07 | 89.11 |  89.8 | 75.09 | 90.48 |  87.7 |    54 | 89.17 |  82.30 |
> | DSIR + neural features         | 83.44 |  88.2 | 89.81 | 70.68 |  90.5 | 87.55 | 52.58 |  88.4 |   81.40 |
>
> > “Retrieval methods: Line 263 references experiments but the results should be included. “
>
> - Line 263 refers to small preliminary data selection tests using BM25 retrieval methods, where we found that when selecting for with AGNews as the target, **out of 6.1M documents retrieved by BM25, there were only 1.8M unique documents (70% were exact duplicates).** We will add these results in the final revision.
> - Since heuristic classification is the method that is used to filter large datasets such as the Pile and the data for GPT3 and PaLM and there is a close similarity between heuristic classification (which uses an inner product score between pretrained word embeddings and a learned vector) and retrieval, we mainly conducted full comparisons against heuristic classification.
>
> > “if a bm25 approach will yield many duplicate documents, why can’t they just be deduplicated? And why does the proposed approach not yield duplicates?”
>
> **Although BM25 retrieved data could be deduplicated, this results in less control over the number of selected examples.** In comparison, DSIR will return exactly the number of examples requested, avoids choosing the exact same document by sampling without replacement, and naturally handles deduplication of repeated documents in the raw data via importance resampling.
>
> In detail, DSIR more gracefully handles duplicates through 2 mechanisms:
> - **1) DSIR avoids sampling the same document by sampling without replacement.** BM25 may select the same document multiple times from different query strings.
> - **2) DSIR naturally decreases the probability to sample an example proportionally to how much it is duplicated in the raw data.** For instance, an example x that is duplicated 1000 times in the raw data will have 1000 times higher probability p(x) under the raw distribution. This reduces the importance weight by a factor of 1000 (since p(x) is in the denominator).
>
> > “ Including decoder/generative experiments may increase the impact of the method”
>
> Due to compute limitations, we weren’t able to pretrain a decoder-only model that is large enough for meaningful few-shot/generative evaluations, but we would like to do so in the future. We thank the reviewer for the suggestion and will add it to the discussion in the final revision.
>
> > “The task shares similarities with active learning, that should include be included somewhere in the related work “
>
>
> **We agree and had already included some active learning works** in the related work (e.g., https://arxiv.org/abs/1901.01151, https://arxiv.org/abs/1708.00489), but we will add more and make the reference to active learning more explicit in the final revision. We thank the reviewer for the suggestion.
>
> > “How are the distributions (q(x) and p(x)) on lines 91/92 computed? … is it simply parameterized by counting n-grams in the respective datasets?”
>
> Yes, they are computed by counting n-gram frequencies. This allows DSIR with n-gram features to be cheap to run on large raw datasets.

---

> > ### Author Response · Authors · 2023-08-14
> > **Further discussion**
> >
> > Dear reviewer, may we ask if you could respond to our comments? In our response, we explore the use of neural features, clarify the comparison to retrieval methods, and addressed other detailed concerns. Please let us know if you have other questions or concerns. Thank you!

---

> > > ### Comment · Reviewer_LSWj · 2023-08-16
> > >
> > > Thanks for the response!
> > >
> > > > Although BM25 retrieved data could be deduplicated, this results in less control over the number of selected examples.
> > >
> > > Why not deduplicate the data before doing retrieval? This is typical (see e.g. https://arxiv.org/pdf/2302.13971.pdf)

---

> > > > ### Author Response · Authors · 2023-08-17
> > > > **Duplication from BM25**
> > > >
> > > > Thanks for the response!
> > > >
> > > > > Why not deduplicate the data before doing retrieval?
> > > >
> > > > To clarify, in the retrieval process, we use BM25 to retrieve the top-k examples (e.g. k=5) for each example in the target dataset and put all the retrieved examples together in a dataset. **The duplication we refer to is from raw documents that show up in the top-k retrieved examples for multiple target examples. This duplication can happen even if we are retrieving from deduplicated data.** To test this, we checked for duplication in document IDs (unique identifier); there were still only 30% unique document IDs in the retrieved dataset, suggesting that the duplication is coming from choosing the same document multiple times, not different documents with the same data.

---

> > > > > ### Author Response · Authors · 2023-08-20
> > > > > **Further discussion**
> > > > >
> > > > > Dear reviewer, please let us know if we've addressed your concerns regarding deduplication above. We are happy to answer any other questions you may have. Thank you!

---

### Decision · Program_Chairs · 2023-09-21

**Decision:**

Accept (poster)

**Comment:**

In sum, reviewers were positive on this paper, with two arguing strongly for acceptance and others closer to the borderline.  One primary concern brought up in the less positive reviews was that the authors only experimented with n-gram based features for data selection.  To address this, the author response included preliminary experiments with dense representations, a helpful data point (although the approach did not show improvements over the ngram approach in the paper).  Incorporating these in the final paper and addressing the other reviewer comments would improve the work.  One additional suggestion: while the related work notes potential shortcomings of Moore-Lewis selection, an experiment evaluating against this classical approach seems like it would strengthen the work.